# $f$-GANs in an Information Geometric Nutshell

**Richard Nock**[†,‡,§]     **Zac Cranko**[‡,†]     **Aditya Krishna Menon**[†,‡]
**Lizhen Qu**[†,‡]     **Robert C. Williamson**[‡,†]
[†]Data61, [‡]the Australian National University and [§]the University of Sydney
{firstname.lastname, aditya.menon, bob.williamson}@data61.csiro.au

## Abstract

Nowozin *et al* showed last year how to extend the GAN *principle* to all $f$-divergences. The approach is elegant but falls short of a full description of the supervised game, and says little about the key player, the generator: for example, what does the generator actually converge to if solving the GAN game means convergence in some space of parameters? How does that provide hints on the generator's design and compare to the flourishing but almost exclusively experimental literature on the subject? In this paper, we unveil a broad class of distributions for which such convergence happens — namely, deformed exponential families, a wide superset of exponential families —. We show that current deep architectures are able to factorize a very large number of such densities using an especially compact design, hence displaying the power of deep architectures and their concinnity in the $f$-GAN game. This result holds given a sufficient condition on *activation functions* — which turns out to be satisfied by popular choices. The key to our results is a variational generalization of an old theorem that relates the KL divergence between regular exponential families and divergences between their natural parameters. We complete this picture with additional results and experimental insights on how these results may be used to ground further improvements of GAN architectures, via (i) a principled design of the activation functions in the generator and (ii) an explicit integration of proper composite losses' link function in the discriminator.

## 1 Introduction

In a recent paper, Nowozin *et al.* [30] showed that the GAN principle [15] can be extended to the variational formulation of all $f$-divergences. In the GAN game, there is an unknown distribution $\mathbb{P}$ which we want to approximate using a parameterised distribution $\mathbb{Q}$. $\mathbb{Q}$ is learned by a **generator** by finding a saddle point of a function which we summarize for now as $f$-GAN$(\mathbb{P}, \mathbb{Q})$, where $f$ is a convex function (see eq. (7) below for its formal expression). A part of the generator's training involves as a subroutine a supervised *adversary* — hence, the saddle point formulation – called **discriminator**, which tries to guess whether randomly generated observations come from $\mathbb{P}$ or $\mathbb{Q}$. Ideally, at the end of this *supervised game*, we want $\mathbb{Q}$ to be close to $\mathbb{P}$, and a good measure of this is the $f$-divergence $I_f(\mathbb{P}\|\mathbb{Q})$, also known as Ali-Silvey distance [1, 12]. Initially, one choice of $f$ was considered [15]. Nowozin *et al.* significantly grounded the game and expanded its scope by showing that for any $f$ convex and suitably defined, then [30, Eq. 4]:

$$\boxed{f\text{-GAN}(\mathbb{P}, \mathbb{Q}) \leq I_f(\mathbb{P}\|\mathbb{Q})} \ . \tag{1}$$

The inequality is an equality if the discriminator is powerful enough. So, solving the $f$-GAN game can give guarantees on how $\mathbb{P}$ and $\mathbb{Q}$ are distant to each other in terms of $f$-divergence. This elegant characterization of the supervised game unfortunately falls short of justifying or elucidating all parameters of the supervised game [30, Section 2.4], and the paper is also silent regarding a key part of the game: the link between distributions in the variational formulation and the *generator*, the

main player which learns a parametric model of a density. In doing so, the $f$-GAN approach and its members remain within an information theoretic framework that relies on divergences between distributions only [30]. In the GAN world at large, this position contrasts with other prominent approaches that explicitly optimize *geometric* distortions between the parameters or support of distributions [6, 14, 16, 21, 22], and raises the problem of connecting the $f$-GAN approach to any sort of information *geometric* optimization. One such information-theoretic/information-geometric identity is well known: The Kullback-Leibler (KL) divergence between two distributions of the *same (regular) exponential family* equals a Bregman divergence $D$ between their natural parameters [2, 4, 7, 9, 35], which we can summarize as:

$$\boxed{I_{f_{\text{KL}}}(\mathbb{P}\|\mathbb{Q}) = D(\boldsymbol{\theta}\|\boldsymbol{\vartheta})} \ . \tag{2}$$

Here, $\boldsymbol{\theta}$ and $\boldsymbol{\vartheta}$ are respectively the natural parameters of $\mathbb{P}$ and $\mathbb{Q}$. Hence, distributions are points on a manifold on the right-hand side, a powerful geometric statement [4]; however, being restricted to KL divergence or "just" exponential families, it certainly falls short of the power to explain the GAN game. To our knowledge, the only generalizations known fall short of the $f$-divergence formulation and are not amenable to the variational GAN formulation [5, Theorem 9], [13, Theorem 3].

**Our first contribution** is such an identity that connects the general $I_f$-divergence formulation in eq. (1) to the general $D$ (Bregman) divergence formulation in eq. (2). We now briefly state it, postponing the details to Section 3:

$$\boxed{f\text{-GAN}(\mathbb{P}, escort(\mathbb{Q})) = D(\boldsymbol{\theta}\|\boldsymbol{\vartheta}) + \text{Penalty}(\mathbb{Q})} \ , \tag{3}$$

for $\mathbb{P}$ and $\mathbb{Q}$ (with respective parameters $\boldsymbol{\theta}$ and $\boldsymbol{\vartheta}$) which happen to lie in a superset of exponential families called *deformed exponential families*, that have received extensive treatment in statistical physics and differential information geometry over the last decade [3, 25]. The right-hand side of eq. (3) is the information geometric part [4], in which $D$ is a Bregman divergence. Therefore, the $f$-GAN problem can be equivalent to a geometric optimization problem [4], like for the Wasserstein GAN and its variants [6]. Notice also that $\mathbb{Q}$ appears in the game in the form of an *escort* [5]. The difference vanish only for exponential families ($escort(\mathbb{Q}) = \mathbb{Q}$, Penalty$(\mathbb{Q}) = 0$ and $f = $ KL).

**Our second contribution** drills down into the information-theoretic and information-geometric parts of (3). In particular, from the former standpoint, we completely specify the parameters of the supervised game, unveiling a key parameter left arbitrary in [30] (explicitly incorporating the link function of proper composite losses [32]). From the latter standpoint, we show that the standard deep generator architecture is powerful at modelling complex escorts of any deformed exponential family, factorising a number of escorts in order of the *total* inner layers' dimensions, and this factorization happens for an especially compact design. This hints on a simple sufficient condition on the activation function to guarantee the escort modelling, and it turns out that this condition is satisfied, exactly or in a limit sense, by most popular activation functions (ELU, ReLU, Softplus, ...). We also provide experiments[1] that display the uplift that can be obtained through a principled design of the activation function (generator), or tuning of the link function (discriminator).

Due to the lack of space, a supplement (SM) provides the proof of the results in the main file and additional experiments. A longer version with a more exhaustive treatment of related results is available [27]. The rest of this paper is as follows. Section § 2 presents definition, § 3 formally presents eq. (3), § 4 derives consequences for deep learning, § 5 completes the supervised game picture of [30], Section § 6 presents experiments and a last Section concludes.

## 2 Definitions

Throughout this paper, the *domain* $\mathcal{X}$ of *observations* is a measurable set. We begin with two important classes of distortion measures, $f$-divergences and Bregman divergences.

**Definition 1** *For any two distributions $\mathbb{P}$ and $\mathbb{Q}$ having respective densities $P$ and $Q$ absolutely continuous with respect to a base measure $\mu$, the $f$-divergence between $\mathbb{P}$ and $\mathbb{Q}$, where $f : \mathbb{R}_+ \to \mathbb{R}$ is convex with $f(1) = 0$, is*

$$I_f(\mathbb{P}\|\mathbb{Q}) \quad \doteq \quad \mathbb{E}_{\mathsf{X}\sim\mathbb{Q}}\left[f\left(\frac{P(\mathsf{X})}{Q(\mathsf{X})}\right)\right] = \int_{\mathcal{X}} Q(\boldsymbol{x}) \cdot f\left(\frac{P(\boldsymbol{x})}{Q(\boldsymbol{x})}\right) \mathrm{d}\mu(\boldsymbol{x}) \ . \tag{4}$$

*For any convex differentiable $\varphi : \mathbb{R}^d \to \mathbb{R}$, the ($\varphi$-)Bregman divergence between $\boldsymbol{\theta}$ and $\boldsymbol{\varrho}$ is:*

$$D_\varphi(\boldsymbol{\theta}\|\boldsymbol{\varrho}) \quad \dot{=} \quad \varphi(\boldsymbol{\theta}) - \varphi(\boldsymbol{\varrho}) - (\boldsymbol{\theta} - \boldsymbol{\varrho})^\top \nabla\varphi(\boldsymbol{\varrho}) \ , \tag{5}$$

*where $\varphi$ is called the generator of the Bregman divergence.*

$f$-divergences are the key distortion measure of information theory, while Bregman divergences are the key distortion measure of information geometry. A distribution $\mathbb{P}$ from a (regular) exponential family with cumulant $C : \Theta \to \mathbb{R}$ and sufficient statistics $\boldsymbol{\phi} : \mathcal{X} \to \mathbb{R}^d$ has density $P_C(\boldsymbol{x}|\boldsymbol{\theta}, \boldsymbol{\phi}) \dot{=} \exp(\boldsymbol{\phi}(\boldsymbol{x})^\top \boldsymbol{\theta} - C(\boldsymbol{\theta}))$, where $\Theta$ is a convex open set, $C$ is convex and ensures normalization on the simplex (we leave implicit the associated dominating measure [3]). A fundamental Theorem ties Bregman divergences and $f$-divergences: when $\mathbb{P}$ and $\mathbb{Q}$ belong to the same exponential family, and denoting their respective densities $P_C(\boldsymbol{x}|\boldsymbol{\theta}, \boldsymbol{\phi})$ and $Q_C(\boldsymbol{x}|\boldsymbol{\vartheta}, \boldsymbol{\phi})$, it holds that $I_{\mathrm{KL}}(\mathbb{P}\|\mathbb{Q}) = D_C(\boldsymbol{\vartheta}\|\boldsymbol{\theta})$. Here, $I_{\mathrm{KL}}$ is Kullback-Leibler (KL) $f$-divergence ($f \dot{=} x \mapsto x \log x$). Remark that the arguments in the Bregman divergence are permuted with respect to those in eq. (2) in the introduction. This also holds if we consider $f_{\mathrm{KL}}$ in eq. (2) to be the Csiszár dual of $f$ [8], namely $f_{\mathrm{KL}} : x \mapsto -\log x$, since in this case $I_{f_{\mathrm{KL}}}(\mathbb{P}\|\mathbb{Q}) = I_{\mathrm{KL}}(\mathbb{Q}\|\mathbb{P}) = D_C(\boldsymbol{\theta}\|\boldsymbol{\vartheta})$. We made this choice in the introduction for the sake of readability in presenting eqs. (1 — 3). We now define generalizations of exponential families, following [5, 13]. Let $\chi : \mathbb{R}_+ \to \mathbb{R}_+$ be non-decreasing [25, Chapter 10]. We define the $\chi$-logarithm, $\log_\chi$, as $\log_\chi(z) \dot{=} \int_1^z \frac{1}{\chi(t)} \mathrm{d}t$. The $\chi$-exponential is $\exp_\chi(z) \dot{=} 1 + \int_0^z \lambda(t)\mathrm{d}t$, where $\lambda$ is defined by $\lambda(\log_\chi(z)) \dot{=} \chi(z)$. In the case where the integrals are improper, we consider the corresponding limit in the argument / integrand.

**Definition 2** *[5] A distribution $\mathbb{P}$ from a $\chi$-exponential family (or deformed exponential family, $\chi$ being implicit) with convex cumulant $C : \Theta \to \mathbb{R}$ and sufficient statistics $\boldsymbol{\phi} : \mathcal{X} \to \mathbb{R}^d$ has density given by $P_{\chi,C}(\boldsymbol{x}|\boldsymbol{\theta}, \boldsymbol{\phi}) \dot{=} \exp_\chi(\boldsymbol{\phi}(\boldsymbol{x})^\top \boldsymbol{\theta} - C(\boldsymbol{\theta}))$, with respect to a dominating measure $\mu$. Here, $\Theta$ is a convex open set and $\boldsymbol{\theta}$ is called the coordinate of $\mathbb{P}$. The **escort density** (or $\chi$-escort) of $P_{\chi,C}$ is*

$$\tilde{P}_{\chi,C} \quad \dot{=} \quad \frac{1}{Z} \cdot \chi(P_{\chi,C}) \ , \quad Z \dot{=} \int_{\mathcal{X}} \chi(P_{\chi,C}(\boldsymbol{x}|\boldsymbol{\theta}, \boldsymbol{\phi}))\mathrm{d}\mu(\boldsymbol{x}) \ . \tag{6}$$

*$Z$ is the escort's normalization constant.*

We leaving implicit the dominating measure and denote $\tilde{\mathbb{P}}$ the escort distribution of $\mathbb{P}$ whose density is given by eq. (6). We shall name $\chi$ the *signature* of the deformed (or $\chi$-)exponential family, and sometimes drop indexes to save readability without ambiguity, noting *e.g.* $\tilde{P}$ for $\tilde{P}_{\chi,C}$. Notice that normalization in the escort is ensured by a simple integration [5, Eq. 7]. For the escort to exist, we require that $Z < \infty$ and therefore $\chi(P)$ is finite almost everywhere. Such a requirement would be satisfied in the GAN game. There is another generalization of regular exponential families, known as *generalized exponential families* [13, 27]. The starting point of our result is the following Theorem, in which the information-theoretic part is not amenable to the variational GAN formulation.

**Theorem 3** *[5][36] for any two $\chi$-exponential distributions $\mathbb{P}$ and $\mathbb{Q}$ with respective densities $P_{\chi,C}, Q_{\chi,C}$ and coordinates $\boldsymbol{\theta}, \boldsymbol{\vartheta}$, $D_C(\boldsymbol{\theta}\|\boldsymbol{\vartheta}) = \mathbb{E}_{\mathsf{X}\sim\tilde{\mathbb{Q}}}[\log_\chi(Q_{\chi,C}(\mathsf{X})) - \log_\chi(P_{\chi,C}(\mathsf{X}))]$.*

We now briefly frame the now popular ($f$-)GAN adversarial learning [15, 30]. We have a true unknown distribution $\mathbb{P}$ over a set of objects, *e.g.* 3D pictures, which we want to learn. In the GAN setting, this is the objective of a *generator*, who learns a distribution $\mathbb{Q}_{\boldsymbol{\theta}}$ parameterized by vector $\boldsymbol{\theta}$. $\mathbb{Q}_{\boldsymbol{\theta}}$ works by passing (the support of) a simple, uninformed distribution, *e.g.* standard Gaussian, through a possibly complex function, *e.g.* a deep net whose parameters are $\boldsymbol{\theta}$ and maps to the support of the objects of interest. Fitting $\mathbb{Q}$ involves an *adversary* (the discriminator) as subroutine, which fits *classifiers*, *e.g.* deep nets, parameterized by $\boldsymbol{\omega}$. The generator's objective is to come up with $\arg\min_{\boldsymbol{\theta}} L_f(\boldsymbol{\theta})$ with $L_f(\boldsymbol{\theta})$ the discriminator's objective:

$$L_f(\boldsymbol{\theta}) \quad \dot{=} \quad \sup_{\boldsymbol{\omega}}\{\mathbb{E}_{\mathsf{X}\sim\mathbb{P}}[T_{\boldsymbol{\omega}}(\mathsf{X})] - \mathbb{E}_{\mathsf{X}\sim\mathbb{Q}_{\boldsymbol{\theta}}}[f^\star(T_{\boldsymbol{\omega}}(\mathsf{X}))]\} \ , \tag{7}$$

where $\star$ is Legendre conjugate [10] and $T_{\boldsymbol{\omega}} : \mathcal{X} \to \mathbb{R}$ integrates the classifier of the discriminator and is therefore parameterized by $\boldsymbol{\omega}$. $L_f$ is a variational approximation to a $f$-divergence [30]; the discriminator's objective is to segregate true ($\mathbb{P}$) from fake ($\mathbb{Q}$) data. The original GAN choice, [15]

$$f_{\mathrm{GAN}}(z) \quad \dot{=} \quad z \log z - (z+1) \log(z+1) + 2\log 2 \tag{8}$$

(the constant ensures $f(1) = 0$) can be replaced by any convex $f$ meeting mild assumptions.

# 3 A variational information geometric identity for the $f$-GAN game

We deliver a series of results that will bring us to formalize eq. (3). First, we define a new set of distortion measures, that we call $KL_\chi$ divergences.

**Definition 4** *For any $\chi$-logarithm and distributions $\mathbb{P}, \mathbb{Q}$ having respective densities $P$ and $Q$ absolutely continuous with respect to base measure $\mu$, the $KL_\chi$ divergence between $\mathbb{P}$ and $\mathbb{Q}$ is defined as $KL_\chi(\mathbb{P}\|\mathbb{Q}) \doteq \mathbb{E}_{\mathsf{X}\sim\mathbb{P}}\left[-\log_\chi(Q(\mathsf{X})/P(\mathsf{X}))\right]$.*

Since $\chi$ is non-decreasing, $-\log_\chi$ is convex and so any $KL_\chi$ divergence is an $f$-divergence. When $\chi(z) \doteq z$, $KL_\chi$ is the KL divergence. In what follows, base measure $\mu$ and absolute continuity are implicit, as well as that $P$ (resp. $Q$) is the density of $\mathbb{P}$ (resp. $\mathbb{Q}$). We now relate $KL_\chi$ divergences vs $f$-divergences. Let $\partial f$ be the subdifferential of convex $f$ and $\mathbb{I}_{P,Q} \doteq [\inf_{\boldsymbol{x}} P(\boldsymbol{x})/Q(\boldsymbol{x}), \sup_{\boldsymbol{x}} P(\boldsymbol{x})/Q(\boldsymbol{x})) \subseteq \mathbb{R}_+$ denote the range of density ratios of $P$ over $Q$. Our first result states that if there is a selection of the subdifferential which is upperbounded on $\mathbb{I}_{P,Q}$, the $f$-divergence $I_f(\mathbb{P}\|\mathbb{Q})$ is equal to a $KL_\chi$ divergence.

**Theorem 5** *Suppose that $\mathbb{P}, \mathbb{Q}$ are such that there exists a selection $\xi \in \partial f$ with $\sup \xi(\mathbb{I}_{P,Q}) < \infty$. Then $\exists \chi : \mathbb{R}_+ \to \mathbb{R}_+$ non decreasing such that $I_f(\mathbb{P}\|\mathbb{Q}) = KL_\chi(\mathbb{Q}\|\mathbb{P})$.*

Theorem 5 essentially covers most if not all relevant GAN cases, as the assumption has to be satisfied in the GAN game for its solution not to be vacuous up to a large extent (eq. (7)). We provide a more complete treatment in the extended version [27]. The proof of Theorem 5 (in SM, Section I) is constructive: it shows how to pick $\chi$ which satisfies all requirements. It brings the following interesting corollary: under mild assumptions on $f$, there exists a $\chi$ that fits for all densities $P$ and $Q$. A prominent example of $f$ that fits is the original GAN choice for which we can pick

$$\chi_{\text{GAN}}(z) \quad \doteq \quad \frac{1}{\log\left(1 + \frac{1}{z}\right)} \ . \tag{9}$$

We now define a slight generalization of $KL_\chi$-divergences and allow for $\chi$ to depend on the choice of the expectation's $\mathsf{X}$, granted that for any of these choices, it will meet the constraints to be $\mathbb{R}_+ \to \mathbb{R}_+$ and also increasing, and therefore define a valid signature. For any $f : \mathcal{X} \to \mathbb{R}_+$, we denote $KL_{\chi_f}(\mathbb{P}\|\mathbb{Q}) \doteq \mathbb{E}_{\mathsf{X}\sim\mathbb{P}}\left[-\log_{\chi_{f(\mathsf{X})}}(Q(\mathsf{X})/P(\mathsf{X}))\right]$, where for any $p \in \mathbb{R}_+$, $\chi_p(t) \doteq \frac{1}{p} \cdot \chi(tp)$. Whenever $f = 1$, we just write $KL_\chi$ as we already did in Definition 4. We note that for any $\boldsymbol{x} \in \mathcal{X}$, $\chi_{f(\boldsymbol{x})}$ is increasing and non negative because of the properties of $\chi$ and $f$, so $\chi_{f(\boldsymbol{x})}(t)$ defines a $\chi$-logarithm. We are ready to state a Theorem that connects $KL_\chi$-divergences and Theorem 3.

**Theorem 6** *Letting $P \doteq P_{\chi,C}$ and $Q \doteq Q_{\chi,C}$ for short in Theorem 3, we have $\mathbb{E}_{\mathsf{X}\sim\tilde{\mathbb{Q}}}[\log_\chi(Q(\mathsf{X})) - \log_\chi(P(\mathsf{X}))] = KL_{\chi_{\tilde{Q}}}(\tilde{\mathbb{Q}}\|\mathbb{P}) - J(\mathbb{Q})$, with $J(\mathbb{Q}) \doteq KL_{\chi_{\tilde{Q}}}(\tilde{\mathbb{Q}}\|\mathbb{Q})$.*

(Proof in SM, Section II) To summarize, we know that under mild assumptions relatively to the GAN game, $f$-divergences coincide with $KL_\chi$ divergences (Theorems 5). We also know from Theorem 6 that $KL_\chi$ divergences quantify the geometric proximity between the coordinates of generalized exponential families (Theorem 3). Hence, finding a geometric (parameter-based) interpretation of the variational $f$-GAN game as described in eq. (7) can be done via a variational formulation of the $KL_\chi$ divergences appearing in Theorem 6. Since penalty $J(\mathbb{Q})$ does not belong to the GAN game (it does not depend on $\mathbb{P}$), it reduces our focus on $KL_{\chi_{\tilde{Q}}}(\tilde{\mathbb{Q}}\|\mathbb{P})$.

**Theorem 7** *$KL_{\chi_{\tilde{Q}}}(\tilde{\mathbb{Q}}\|\mathbb{P})$ admits the variational formulation*

$$KL_{\chi_{\tilde{Q}}}(\tilde{\mathbb{Q}}\|\mathbb{P}) \quad = \quad \sup_{T \in \overline{\mathbb{R}_{++}}^{\mathcal{X}}} \left\{ \mathbb{E}_{\mathsf{X}\sim\mathbb{P}}[T(\mathsf{X})] - \mathbb{E}_{\mathsf{X}\sim\tilde{\mathbb{Q}}}[(-\log_{\chi_{\tilde{Q}}})^\star(T(\mathsf{X}))] \right\} \ , \tag{10}$$

*with $\overline{\mathbb{R}_{++}} \doteq \mathbb{R}\backslash\mathbb{R}_{++}$. Furthermore, letting $Z$ denoting the normalization constant of the $\chi$-escort of $Q$, the optimum $T^* : \mathcal{X} \to \overline{\mathbb{R}_{++}}$ to eq. (10) is $T^*(\boldsymbol{x}) = -(1/Z) \cdot (\chi(Q(\boldsymbol{x}))/\chi(P(\boldsymbol{x})))$.*

(Proof in SM, Section III) Hence, the variational $f$-GAN formulation can be captured in an information-geometric framework by the following identity using Theorems 3, 5, 7.

**Corollary 8** *(the variational information-geometric $f$-GAN identity) Using notations from Theorems 6, 7 and letting $\boldsymbol{\theta}$ (resp. $\boldsymbol{\vartheta}$) denote the coordinate of $\mathbb{P}$ (resp. $\mathbb{Q}$), we have:*

$$\boxed{\sup_{T\in\overline{\mathbb{R}_{++}}^{\mathfrak{X}}}\left\{\mathbb{E}_{\mathsf{X}\sim\mathbb{P}}[T(\mathsf{X})]-\mathbb{E}_{\mathsf{X}\sim\tilde{\mathbb{Q}}}[(-\log_{\chi_{\tilde{Q}}})^{\star}(T(\mathsf{X}))]\right\}=D_C(\boldsymbol{\theta}\|\boldsymbol{\vartheta})+J(\mathbb{Q})}\ . \tag{11}$$

We shall also name for short *vig-$f$-GAN* the identity in eq. (11). We note that we can drill down further the identity, expressing in particular the Legendre conjugate $(-\log_{\chi_{\tilde{Q}}})^{\star}$ with an equivalent "dual" (negative) $\chi$-logarithm in the variational problem [27]. The left hand-side of Eq. (11) has the exact same overall shape as the variational objective of [30, Eqs 2, 6]. However, it tells the formal story of GANs in significantly greater details, in particular for what concerns the generator. For example, eq. (11) yields a new characterization of the generators' convergence: because $D_C$ is a Bregman divergence, it satisfies the identity of the indiscernibles. So, solving the $f$-GAN game [30] can guarantees convergence in the parameter space ($\boldsymbol{\vartheta}$ vs $\boldsymbol{\theta}$). In the realm of GAN applications, it makes sense to consider that $\mathbb{P}$ (the true distribution) can be extremely complex. Therefore, even when deformed exponential families are significantly more expressive than regular exponential families [25], extra care should be put before arguing that complex applications comply with such a geometric convergence in the parameter space. One way to circumvent this problem is to build distributions in $\mathbb{Q}$ that factorize many deformed exponential families. This is one strong point of deep architectures that we shall prove next.

## 4 Deep architectures in the vig-$f$-GAN game

In the GAN game, distribution $\mathbb{Q}$ in eq. (11) is built by the generator (call it $\mathbb{Q}_g$), by passing the support of a simple distribution (*e.g.* uniform, standard Gaussian), $\mathbb{Q}_{\text{in}}$, through a series of non-linear transformations. Letting $Q_{\text{in}}$ denote the corresponding density, we now compute $Q_g$. Our generator $\boldsymbol{g}:\mathfrak{X}\to\mathbb{R}^d$ consists of two parts: a deep part and a last layer. The deep part is, given some $L\in\mathbb{N}$, the computation of a non-linear transformation $\boldsymbol{\phi}_L:\mathfrak{X}\to\mathbb{R}^{d_L}$ as

$$\mathbb{R}^{d_l}\ni\boldsymbol{\phi}_l(\boldsymbol{x})\ \doteq\ \boldsymbol{v}(\mathrm{W}_l\boldsymbol{\phi}_{l-1}(\boldsymbol{x})+\boldsymbol{b}_l)\ ,\forall l\in\{1,2,...,L\}\ , \tag{12}$$
$$\boldsymbol{\phi}_0(\boldsymbol{x})\ \doteq\ \boldsymbol{x}\in\mathfrak{X}\ . \tag{13}$$

$\boldsymbol{v}$ is a function computed coordinate-wise, such as (leaky) ReLUs, ELUs [11, 17, 23, 24], $\mathrm{W}_l\in\mathbb{R}^{d_l\times d_{l-1}}$, $\boldsymbol{b}_l\in\mathbb{R}^{d_l}$. The last layer computes the generator's output from $\boldsymbol{\phi}_L$: $\boldsymbol{g}(\boldsymbol{x})\doteq\boldsymbol{v}_{\text{OUT}}(\Gamma\boldsymbol{\phi}_L(\boldsymbol{x})+\boldsymbol{\beta})$, with $\Gamma\in\mathbb{R}^{d\times d_L},\boldsymbol{\beta}\in\mathbb{R}^d$; in general, $v_{\text{OUT}}\neq v$ and $v_{\text{OUT}}$ fits the output to the domain at hand, ranging from linear [6, 20] to non-linear functions like $\tanh$ [30]. Our generator captures the high-level features of some state of the art generative approaches [31, 37].

To carry our analysis, we make the assumption that the network is reversible, which is going to require that $v_{\text{OUT}},\Gamma,\mathrm{W}_l$ ($l\in\{1,2,...,L\}$) are invertible. We note that popular examples can be invertible (*e.g.* DCGAN, if we use $\mu$-ReLU, dimensions match and fractional-strided convolutions are invertible). At this reasonable price, we get in closed form the generator's density and it shows the following: for any continuous signature $\chi_{\text{net}}$, there exists an activation function $v$ such that the deep part in the network factors as escorts for the $\chi_{\text{net}}$-exponential family. Let $\mathbf{1}_i$ denote the $i^{th}$ canonical basis vector.

**Theorem 9** $\forall v_{\text{OUT}},\Gamma,\mathrm{W}_l$ *invertible ($l\in\{1,2,...,L\}$), for any continuous signature $\chi_{net}$, there exists activation $v$ and $\boldsymbol{b}_l\in\mathbb{R}^d$ ($\forall l\in\{1,2,...,L\}$) such that for any output $\boldsymbol{z}$, letting $\boldsymbol{x}\doteq\boldsymbol{g}^{-1}(\boldsymbol{z})$, $Q_g(\boldsymbol{z})$ factorizes as $Q_g(\boldsymbol{z})=(Q_{in}(\boldsymbol{x})/\tilde{Q}_{deep}(\boldsymbol{x}))\cdot 1/(H_{out}(\boldsymbol{x})\cdot Z_{net})$, with $Z_{net}>0$ a constant, $H_{out}(\boldsymbol{x})\doteq\prod_{i=1}^d|v'_{\text{OUT}}(\boldsymbol{\gamma}_i^\top\boldsymbol{\phi}_L(\boldsymbol{x})+\beta_i)|$, $\boldsymbol{\gamma}_i\doteq\Gamma^\top\mathbf{1}_i$, and (letting $\boldsymbol{w}_{l,i}\doteq\mathrm{W}_l^\top\mathbf{1}_i$):*

$$\tilde{Q}_{deep}(\boldsymbol{x})\ \doteq\ \prod_{l=1}^L\prod_{i=1}^d\tilde{P}_{\chi_{net},b_{l,i}}(\boldsymbol{x}|\boldsymbol{w}_{l,i},\boldsymbol{\phi}_{l-1})\ . \tag{14}$$

| Name | $v(z)$ | $\chi(z)$ |
|---|---|---|
| ReLU$^{(\S)}$ | $\max\{0, z\}$ | $1_{z>0}$ |
| Leaky-ReLU$^{(\dagger)}$ | $\begin{cases} z & \text{if} \quad z > 0 \\ \epsilon z & \text{if} \quad z \le 0 \end{cases}$ | $\begin{cases} 1 & \text{if} \quad z > -\delta \\ \frac{1}{\epsilon} & \text{if} \quad z \le -\delta \end{cases}$ |
| $(\alpha, \beta)$-ELU$^{(\heartsuit)}$ | $\begin{cases} \beta z & \text{if} \quad z > 0 \\ \alpha(\exp(z) - 1) & \text{if} \quad z \le 0 \end{cases}$ | $\begin{cases} \beta & \text{if} \quad z > \alpha \\ z & \text{if} \quad z \le \alpha \end{cases}$ |
| prop-$\tau^{(\clubsuit)}$ | $k + \frac{\tau^\star(z)}{\tau^\star(0)}$ | $\frac{\tau'^{-1} \circ (\tau^\star)^{-1}(\tau^\star(0)z)}{\tau^\star(0)}$ |
| Softplus$^{(\diamond)}$ | $k + \log_2(1 + \exp(z))$ | $\frac{1}{\log 2} \cdot \left(1 - 2^{-z}\right)$ |
| $\mu$-ReLU$^{(\spadesuit)}$ | $k + \frac{z + \sqrt{(1-\mu)^2 + z^2}}{2}$ | $\frac{4z^2}{(1-\mu)^2 + 4z^2}$ |
| LSU | $k + \begin{cases} 0 & \text{if} \quad z < -1 \\ (1+z)^2 & \text{if} \quad z \in [-1, 1] \\ 4z & \text{if} \quad z > 1 \end{cases}$ | $\begin{cases} 2\sqrt{z} & \text{if} \quad z < 4 \\ 4 & \text{if} \quad z > 4 \end{cases}$ |

Table 1: Some strongly/weakly admissible couples $(v, \chi)$. $(\S)$ : 1. is the indicator function; $(\dagger)$ : $\delta \le 0, 0 < \epsilon \le 1$ and $\mathrm{dom}(v) = [\delta/\epsilon, +\infty)$. $(\heartsuit)$ : $\beta \ge \alpha > 0$; $(\clubsuit)$ : $\star$ is Legendre conjugate; $(\spadesuit)$ : $\mu \in [0, 1)$. Shaded: prop-$\tau$ activations; $k$ is a constant (*e.g.* such that $v(0) = 0$) (see text).

(Proof in SM, Section IV) The relationship between the inner layers of a deep net and deformed exponential families (Definition 2) follows from the theorem: rows in W$_l$s define coordinates, $\phi_l$ define "deep" sufficient statistics, $\boldsymbol{b}_l$ are cumulants and the crucial part, the $\chi$-family, is given by the activation function $v$. Notice also that the $\boldsymbol{b}_l$s are learned, and so the deformed exponential families' normalization is in fact *learned* and not specified. We see that $\tilde{Q}_{\mathrm{deep}}$ factors escorts, and in number. What is notable is the compactness achieved by the deep representation: the total dimension of all deep sufficient statistics in $\tilde{Q}_{\mathrm{deep}}$ (eq. (14)) is $L \cdot d$. To handle this, a shallow net with a single inner layer would require a matrix W of space $\Omega(L^2 \cdot d^2)$. The deep net $\boldsymbol{g}$ requires only $O(L \cdot d^2)$ space to store all W$_l$s. The proof of Theorem 9 is constructive: it builds $v$ as a function of $\chi$. In fact, the proof also shows how to build $\chi$ *from* the activation function $v$ in such a way that $\tilde{Q}_{\mathrm{deep}}$ factors $\chi$-escorts. The following Lemma essentially says that this is possible for all *strongly admissible* activations $v$.

**Definition 10** *Activation function $v$ is **strongly admissible** iff $\mathrm{dom}(v) \cap \overline{\mathbb{R}_+} \ne \emptyset$ and $v$ is $C^1$, lowerbounded, strictly increasing and convex. $v$ is **weakly admissible** iff for any $\epsilon > 0$, there exists $v_\epsilon$ strongly admissible such that $||v - v_\epsilon||_{L_1} < \epsilon$, where $||f||_{L_1} \doteq \int |f(t)| \mathrm{d}t$.*

**Lemma 11** *The following holds: (i) for any strongly admissible $v$, there exists signature $\chi$ such that Theorem 9 holds; (ii) $(\gamma, \gamma)$-ELU (for any $\gamma > 0$), Softplus are strongly admissible. ReLU is weakly admissible.*

(proof in SM, Section V) The proof uses a trick for ReLU which can easily be repeated for $(\alpha, \beta)$-ELU, and for leaky-ReLU, with the constraint that the domain has to be lowerbounded. Table 1 provides some examples of strongly / weakly admissible activations. It includes a wide class of so-called "prop-$\tau$ activations", where $\tau$ is negative a concave entropy, defined on $[0, 1]$ and symmetric around $1/2$ [29]. This concludes our treatment of the information geometric part of the vig-$f$-GAN identity. We now complete it with a treatment of its information-theoretic part.

## 5  A complete proper loss picture of the supervised GAN game

In their generalization of the GAN objective, Nowozin *et al.* [30] leave untold a key part of the supervised game: they split in eq. (7) the discriminator's contribution in two, $T_{\boldsymbol{\omega}} = g_f \circ V_{\boldsymbol{\omega}}$, where $V_{\boldsymbol{\omega}} : \mathcal{X} \to \mathbb{R}$ is the actual discriminator, and $g_f$ is essentially a technical constraint to ensure that $V_{\boldsymbol{\omega}}(.)$ is in the domain of $f^\star$. They leave the choice of $g_f$ "somewhat arbitrary" [30, Section 2.4]. We now show that if one wants the supervised loss to have the desirable property to be *proper composite* [32][2], then $g_f$ is not arbitrary. We proceed in three steps, first unveiling a broad class of *proper f-GANs* that deal with this property. The initial motivation of eq. (7) was that the inner maximisation may be seen as the $f$-divergence between $\mathbb{P}$ and $\mathbb{Q}_{\boldsymbol{\theta}}$ [26], $L_f(\boldsymbol{\theta}) = I_f(\mathbb{P}\|\mathbb{Q}_{\boldsymbol{\theta}})$. In fact, this variational

representation of an $f$-divergence holds more generally: by [33, Theorem 9], we know that for any convex $f$, and invertible *link function* $\Psi\colon (0,1) \to \mathbb{R}$, we have:

$$\inf_{T\colon \mathcal{X}\to\mathbb{R}} \mathbb{E}_{(\mathsf{X},\mathsf{Y})\sim\mathbb{D}}\left[\ell_\Psi(\mathsf{Y}, T(\mathsf{X}))\right] = -\frac{1}{2}\cdot I_f(\mathbb{P}\,\|\,\mathbb{Q}) \tag{15}$$

where $\mathbb{D}$ is the distribution over (observations $\times$ {fake, real}) and the loss function $\ell_\Psi$ is defined by:

$$\ell_\Psi(+1, z) \doteq -f'\left(\frac{\Psi^{-1}(z)}{1-\Psi^{-1}(z)}\right) \quad ; \quad \ell_\Psi(-1, z) \doteq f^\star\left(f'\left(\frac{\Psi^{-1}(z)}{1-\Psi^{-1}(z)}\right)\right) \;, \tag{16}$$

assuming $f$ differentiable. Note now that picking $\Psi(z) = f'(z/(1-z))$ with $z \doteq T(\boldsymbol{x})$ and simplifying eq. (15) with $\mathbb{P}[\mathsf{Y}=\text{fake}] = \mathbb{P}[\mathsf{Y}=\text{real}] = 1/2$ in the GAN game yields eq. (7). For other link functions, however, we get an equally valid class of losses whose optimisation will yield a meaningful estimate of the $f$-divergence. The losses of eq. (16) belong to the class of *proper composite losses* with *link function* $\Psi$ [32]. Thus (omitting parameters $\boldsymbol{\theta}, \boldsymbol{\omega}$), we rephrase eq. (7) and refer to the *proper $f$-GAN* formulation as $\inf_\mathbb{Q} L_\Psi(\mathbb{Q})$ with ($\ell$ is as per eq. (16)):

$$L_\Psi(\mathbb{Q}) \doteq \sup_{T\colon \mathcal{X}\to\mathbb{R}}\left\{\mathbb{E}_{\mathsf{X}\sim\mathbb{P}}\left[-\ell_\Psi(+1, T(\mathsf{X}))\right] + \mathbb{E}_{\mathsf{X}\sim\mathbb{Q}}\left[-\ell_\Psi(-1, T(\mathsf{X}))\right]\right\} \;. \tag{17}$$

Note also that it is trivial to start from a suitable proper composite loss, and derive the corresponding generator $f$ for the $f$-divergence as per eq. (15). Finally, our proper composite loss view of the $f$-GAN game allows us to elicitate $g_f$ in [30]: it is the composition of $f'$ and $\Psi$ in eq. (16). The use of proper composite losses as part of the supervised GAN formulation sheds further light on another aspect the game: the connection between the value of the optimal discriminator, and the density ratio between the generator and discriminator distributions. Instead of the optimal $T^*(\boldsymbol{x}) = f'(P(\boldsymbol{x})/Q(\boldsymbol{x}))$ for eq. (7) [30, Eq. 5], we now have with the more general eq. (17) the result $T^*(\boldsymbol{x}) = \Psi((1 + Q(\boldsymbol{x})/P(\boldsymbol{x}))^{-1})$. We now show that proper $f$-GANs can easily be adapted to eq. (11). We let $\chi^\bullet(t) \doteq 1/\chi^{-1}(1/t)$.

**Theorem 12** *For any $\chi$, define $\ell_{\boldsymbol{x}}(-1, z) \doteq -\log_{(\chi^\bullet)_{\frac{1}{\tilde{Q}(\boldsymbol{x})}}}(-z)$, and let $\ell(+1, z) \doteq -z$. Then $L_\Psi(\mathbb{Q})$ in eq. (17) equals eq. (11). Its link in eq. (17) is $\Psi_{\boldsymbol{x}}(z) = -1/\chi_{\tilde{Q}(\boldsymbol{x})}(z/(1-z))$.*

(Proof in SM, Section VI) Hence, in the proper composite view of the vig-$f$-GAN identity, the generator rules over the supervised game: it tempers with both the link function and the loss — but only for fake examples. Notice also that when $z = -1$, the fake examples loss satisfies $\ell_{\boldsymbol{x}}(-1, -1) = 0$ regardless of $\boldsymbol{x}$ by definition of the $\chi$-logarithm.

## 6 Experiments

Two of our theoretical contributions are (A) the fact that on the *generator*'s side, there exists numerous activation functions $v$ that comply with the design of its density as factoring escorts (Lemma 11), and (B) the fact that on the *discriminator*'s side, the so-called output activation function $g_f$ of [30] aggregates in fact two components of proper composite losses, one of which, the link function $\Psi$, should be a fine knob to operate (Theorem 12). We have tested these two possibilities with the idea that an experimental validation should provide substantial ground to be competitive with mainstream approaches, leaving space for a finer tuning in specific applications. Also, in order not to mix their effects, we have treated (A) and (B) separately.

*Architectures and datasets* — We provide in SM (Section VI) the detail of all experiments. To summarize, we consider two architectures in our experiments: DCGAN [31] and the multilayer feedforward network (MLP) used in [30]. Our datasets are MNIST [19] and LSUN tower category [38].

*Comparison of varying activations in the generator (A)* — We have compared $\mu$-ReLUs with varying $\mu$ in $[0, 0.1, ..., 1]$ (hence, we include ReLU as a baseline for $\mu = 1$), the Softplus and the Least Square Unit (LSU, Table 1) activation (Figure 1). For each choice of the activation function, all inner layers of the generator use the same activation function. We evaluate the activation functions by using both DCGAN and the MLP used in [30] as the architectures. As training divergence, we adopt both GAN [15] and Wasserstein GAN (WGAN, [6]). Results are shown in Figure 1 (left).

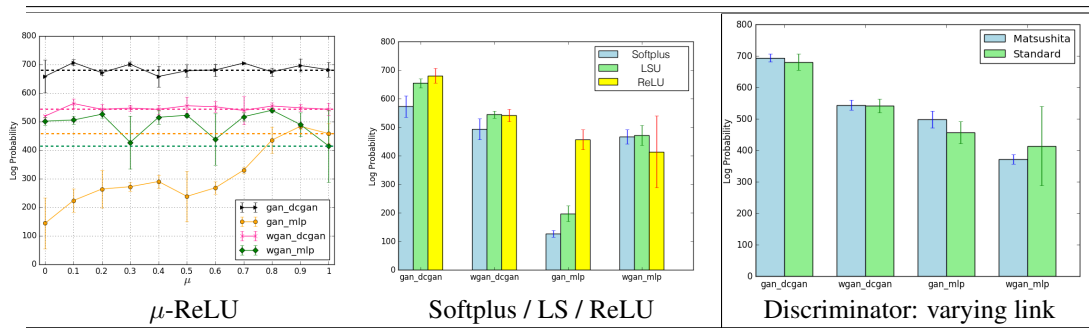

| | | |
|:---:|:---:|:---:|
| $\mu$-ReLU | Softplus / LS / ReLU | Discriminator: varying link |

Figure 1: Summary of our results on MNIST, on experiment A (left+center) and B (right). *Left*: comparison of different values of $\mu$ for the $\mu$-ReLU activation in the generator (ReLU = 1-ReLU, see text). Thicker horizontal dashed lines present the ReLU average baseline: for each color, points above the baselines represent values of $\mu$ for which ReLU is beaten on average. *Center*: comparison of different activations in the generator, for the same architectures as in the left plot. *Right*: comparison of different link function in the discriminator (see text, best viewed in color).

Three behaviours emerge when varying $\mu$: either it is globally equivalent to ReLU (GAN DCGAN) but with local variations that can be better ($\mu = 0.7$) or worse ($\mu = 0$), or it is almost consistently better than ReLU (WGAN MLP) or worse (GAN MLP). The best results were obtained for GAN DCGAN, and we note that the ReLU baseline was essentially beaten for values of $\mu$ yielding smaller variance, and hence yielding smaller uncertainty in the results. The comparison between different activation functions (Figure 1, center) reveals that ($\mu$-)ReLU performs overall the best, yet with some variations among architectures. We note in particular that, in the same way as for the comparisons intra $\mu$-ReLU (Figure 1, left), ReLU performs relatively worse than the other criteria for WGAN MLP, indicating that there may be different best fit activations for different architectures, which is good news. Visual results on LSUN (SM, Table A5) also display the quality of results when changing the $\mu$-ReLU activation.

*Comparison of varying link functions in the discriminator (B)* — We have compared the replacement of the sigmoid function by a link which corresponds to the entropy which is theoretically optimal in boosting algorithms, Matsushita entropy [18, 28], for which $\Psi_{\mathrm{MAT}}(z) \doteq (1/2) \cdot (1 + z/\sqrt{1 + z^2})$. Figure 1 (right) displays the comparison Matsushita vs "standard" (more specifically, we use sigmoid in the case of GAN [30], and none in the case of WGAN to follow current implementations [6]). We evaluate with both DCGAN and MLP on MNIST (same hyperparameters as for generators, ReLU activation for all hidden layer activation of generators). Experiments tend to display that tuning the link may indeed bring additional uplift: for GANs, Matsushita is indeed better than the sigmoid link for both DCGAN and MLP, while it remains very competitive with the no-link (or equivalently an identity link) of WGAN, at least for DCGAN.

## 7 Conclusion

It is hard to exaggerate the success of GAN approaches in modelling complex domains, and with their success comes an increasing need for a rigorous theoretical understanding [34]. In this paper, we complete the supervised understanding of the generalization of GANs introduced in [30], and provide a theoretical background to understand its unsupervised part, showing in particular how deep architectures can be powerful at tackling the generative part of the game. Experiments display that the tools we develop may help to improve further the state of the art.

## 8 Acknowledgments

The authors thank the reviewers, Shun-ichi Amari, Giorgio Patrini and Frank Nielsen for numerous comments.

## Footnotes

[1]The code used for our experiments is available through https://github.com/qulizhen/fgan_info_geometric

[2]informally, Bayes rule realizes the optimum and the loss accommodates for any real valued predictor.

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
