[Supplementary Material]

# $f$-GANs in an Information Geometric Nutshell
## — Supplementary Material —

### Abstract

This is the Supplementary Material to Paper "$f$-GANs in an Information Geometric Nutshell" by R. Nock, Z. Cranko, A-K. Menon, L. Qu and and R.-C. Williamson. Theorems and Lemmata are numbered with letters (A, B, ...) to make a clear difference with the main file numbering.

# Table of contents

# Supplementary material on proofs and formal results

# I  Proof of Theorem 5

Our basis for the proof of the Theorem is the following Lemma.

**Lemma A** *[10, Proposition 1.6.1] Let $f : I \to \mathbb{R}$ be continuous convex and let $\xi : I \to \mathbb{R}$ such that $\xi(z) \in \partial f(z), \forall z \in \mathrm{int} I$. Then for any $a < b$ in $I$, it holds that:*

$$f(b) = f(a) + \int_a^b \xi(t)\mathrm{d}t \ . \tag{1}$$

Suppose that $b < a$. Then Lemma A says that we have $f(a) = f(b) + \int_b^a \xi(t)\mathrm{d}t$, that is, after re-ordering, $f(b) = f(a) - \int_b^a \xi(t)\mathrm{d}t = f(a) + \int_a^b \xi(t)\mathrm{d}t$, so in fact the requested ordering between the integral's bounds can be removed. Also, we can suppose that the integral may not be proper, in which case we compute it as a limit of a proper integral for which Lemma A therefore holds.

We now prove Theorem 5. Suppose there exists $M \in \mathbb{R}$ such that $\sup \xi(\mathbb{I}_{P,Q}) \le M$, for some $\partial f \ni \xi : \mathrm{int} \, \mathrm{dom}(f) \to \mathbb{R}$. For any constants $k$, letting $f_k(z) \doteq f(z) - k(z-1)$, which is convex since $f$ is, we note that

$$
\begin{aligned}
\mathbb{E}_{\mathsf{X} \sim \mathbb{Q}} \left[ f_k \left( \frac{P(\mathsf{X})}{Q(\mathsf{X})} \right) \right] &= \mathbb{E}_{\mathsf{X} \sim \mathbb{Q}} \left[ f \left( \frac{P(\mathsf{X})}{Q(\mathsf{X})} \right) \right] - k \cdot \mathbb{E}_{\mathsf{X} \sim Q} \left[ \frac{P(\mathsf{X})}{Q(\mathsf{X})} - 1 \right] \\
&= \mathbb{E}_{\mathsf{X} \sim \mathbb{Q}} \left[ f \left( \frac{P(\mathsf{X})}{Q(\mathsf{X})} \right) \right] - k \cdot \left( \int P(\mathsf{X})\mathrm{d}\mu(\mathsf{X}) - \int Q(\mathsf{X})\mathrm{d}\mu(\mathsf{X}) \right) \\
&= \mathbb{E}_{\mathsf{X} \sim \mathbb{Q}} \left[ f \left( \frac{P(\mathsf{X})}{Q(\mathsf{X})} \right) \right] \ .
\end{aligned}
\tag{2}
$$

Let $\xi_k \doteq \xi - k \in \partial f_k$. Since $f_k$ is convex continuous, it follows from [10, Proposition 1.6.1] (Lemma A) that:

$$
\begin{aligned}
f_k \left( \frac{P(\boldsymbol{x})}{Q(\boldsymbol{x})} \right) &= f_k(1) + \lim_{\rho \to \frac{P(\boldsymbol{x})}{Q(\boldsymbol{x})}} \int_1^\rho \xi_k(t)\mathrm{d}t \\
&= - \lim_{\rho \to \frac{P(\boldsymbol{x})}{Q(\boldsymbol{x})}} \int_1^\rho (-\xi(t)+k)\mathrm{d}t \ .
\end{aligned}
\tag{3}
$$

The second identity comes from the assumption that $f(1) = 0 = f_k(1)$. The limit appears to cope with a subdifferential that would diverge around a density ratio. Fix some constant $\epsilon > 0$ and let

$$
\chi(t) = \left\{
\begin{array}{cl}
\frac{1}{-\xi(t)+M+\epsilon} & \text{if} \quad t < \sup \mathbb{I}_{P,Q} \\
\frac{1}{\epsilon} & \text{if} \quad t \ge \sup \mathbb{I}_{P,Q}
\end{array}
\right. ,
\tag{4}
$$

which, since $\sup \xi(\mathbb{I}_{P,Q}) \leq M$, guarantees $\chi \geq 0$ and $\chi$ is also increasing since $\xi$ is increasing ($f$ is convex). We then check, using eqs. (2) and (4) that:

$$
\begin{aligned}
KL_\chi(\mathbb{Q}\|\mathbb{P}) &= \mathbb{E}_{\mathsf{X}\sim\mathbb{Q}}\left[-\log_\chi\left(\frac{P(\mathsf{X})}{Q(\mathsf{X})}\right)\right] \\
&= \mathbb{E}_{\mathsf{X}\sim\mathbb{Q}}\left[-\lim_{\rho\to\frac{P(\mathsf{X})}{Q(\mathsf{X})}}\int_1^\rho \frac{1}{\chi(t)}\mathrm{d}t\right] \\
&= \mathbb{E}_{\mathsf{X}\sim\mathbb{Q}}\left[-\lim_{\rho\to\frac{P(\mathsf{X})}{Q(\mathsf{X})}}\int_1^\rho (-\xi(t) + M + \epsilon)\mathrm{d}t\right] \\
&= \mathbb{E}_{\mathsf{X}\sim\mathbb{Q}}\left[f_{M+\epsilon}\left(\frac{P(\mathsf{X})}{Q(\mathsf{X})}\right)\right] \\
&= \mathbb{E}_{\mathsf{X}\sim\mathbb{Q}}\left[f\left(\frac{P(\mathsf{X})}{Q(\mathsf{X})}\right)\right] = I_f(\mathbb{P}\|\mathbb{Q}) \ . \qquad (5)
\end{aligned}
$$

This ends the proof of Theorem 5.

## II   Proof of Theorem 6

We have

$$
\begin{aligned}
\mathbb{E}_{\mathsf{X}\sim\tilde{\mathbb{Q}}}&[-(\log_\chi(P(\mathsf{X})) - \log_\chi(Q(\mathsf{X})))] \\
&= \mathbb{E}_{\mathsf{X}\sim\tilde{\mathbb{Q}}}[-(\log_\chi(P(\mathsf{X})) - \log_\chi(\tilde{Q}(\mathsf{X})))] + \mathbb{E}_{\mathsf{X}\sim\tilde{\mathbb{Q}}}[-(\log_\chi(\tilde{Q}(\mathsf{X})) - \log_\chi(Q(\mathsf{X})))] \\
&= \mathbb{E}_{\mathsf{X}\sim\tilde{\mathbb{Q}}}[-(\log_\chi(P(\mathsf{X})) - \log_\chi(\tilde{Q}(\mathsf{X})))] - \mathbb{E}_{\mathsf{X}\sim\tilde{\mathbb{Q}}}[-(\log_\chi(Q(\mathsf{X})) - \log_\chi(\tilde{Q}(\mathsf{X})))] \ .
\end{aligned}
$$

Consider some fixed $\boldsymbol{x} \in \mathfrak{X}$. We have

$$
\begin{aligned}
\log_\chi(P(\boldsymbol{x})) - \log_\chi(\tilde{Q}(\boldsymbol{x})) &= \int_1^{P(\boldsymbol{x})} \frac{1}{\chi(t)} \cdot \mathrm{d}t - \int_1^{\tilde{Q}(\boldsymbol{x})} \frac{1}{\chi(t)} \cdot \mathrm{d}t \\
&= \int_{\tilde{Q}(\boldsymbol{x})}^{P(\boldsymbol{x})} \frac{1}{\chi(t)} \cdot \mathrm{d}t \\
&= \int_1^{\frac{P(\boldsymbol{x})}{\tilde{Q}(\boldsymbol{x})}} \frac{\tilde{Q}(\boldsymbol{x})}{\chi(t\tilde{Q}(\boldsymbol{x}))} \cdot \mathrm{d}t \\
&= \int_1^{\frac{P(\boldsymbol{x})}{\tilde{Q}(\boldsymbol{x})}} \frac{1}{\chi_{\tilde{Q}(\boldsymbol{x})}(t)} \cdot \mathrm{d}t \\
&= \log_{\chi_{\tilde{Q}(\boldsymbol{x})}}\left(\frac{P(\boldsymbol{x})}{\tilde{Q}(\boldsymbol{x})}\right) \ , \qquad (6)
\end{aligned}
$$

with

$$
\chi_{\tilde{Q}(\boldsymbol{x})}(t) \ \doteq \ \frac{1}{\tilde{Q}(\boldsymbol{x})} \cdot \chi(t\tilde{Q}(\boldsymbol{x})) \ . \qquad (7)
$$

To cope with the case where any of the integrals is improper, we derive the limit expression:

$$(\log_\chi(P(\boldsymbol{x})) - \log_\chi(\tilde{Q}(\boldsymbol{x}))) \;=\; \lim_{(p,q)\to(P(\boldsymbol{x}),\tilde{Q}(\boldsymbol{x}))} \log_{\chi_q}\left(\frac{p}{q}\right) \;, \tag{8}$$

so we get in all cases,

$$\mathbb{E}_{\mathsf{X}\sim\tilde{\mathbb{Q}}}[-(\log_\chi(P(\mathsf{X})) - \log_\chi(\tilde{Q}(\mathsf{X})))] \;=\; KL_{\chi_{\tilde{Q}}}(\tilde{Q}\|P) \;. \tag{9}$$

We also note that

$$\log_\chi(Q(\mathsf{X})) - \log_\chi(\tilde{Q}(\mathsf{X})) \;=\; \lim_{(q,q')\to(Q(\boldsymbol{x}),\tilde{Q}(\boldsymbol{x}))} \log_{\chi_{q'}}\left(\frac{q}{q'}\right)$$

$$=\; \log_{\chi_{\tilde{Q}(\boldsymbol{x})}}\left(\frac{Q(\boldsymbol{x})}{\tilde{Q}(\boldsymbol{x})}\right) \tag{10}$$

(if the limit exists) so we get

$$\mathbb{E}_{\mathsf{X}\sim\tilde{\mathbb{Q}}}[-(\log_\chi(P(\mathsf{X})) - \log_\chi(\tilde{Q}(\mathsf{X})))] \;=\; KL_{\chi_{\tilde{Q}}}(\tilde{Q}\|P) \;, \tag{11}$$

$$\mathbb{E}_{\mathsf{X}\sim\tilde{Q}}[-(\log_\chi(Q(\mathsf{X})) - \log_\chi(\tilde{Q}(\mathsf{X})))] \;=\; KL_{\chi_{\tilde{Q}}}(\tilde{Q}\|Q) \;, \tag{12}$$

and

$$\mathbb{E}_{\mathsf{X}\sim\tilde{\mathbb{Q}}}[\log_\chi(Q(\mathsf{X})) - \log_\chi(P(\mathsf{X}))] \;=\; KL_{\chi_{\tilde{Q}}}(\tilde{\mathbb{Q}}\|\mathbb{P}) - KL_{\chi_{\tilde{Q}}}(\tilde{\mathbb{Q}}\|\mathbb{Q}) \;, \tag{13}$$

as claimed.

# III   Proof of Theorem 7

Let us denote $\mathcal{F}_{\tilde{Q}} \subseteq \mathbb{R}^{\mathcal{X}}$ denote the subset of functions : $\mathcal{X} \to \mathbb{R}$ whose values are constrained as follows:

$$\mathcal{F}_{\tilde{Q}} \;\doteq\; \left\{ T \in \mathbb{R}^{\mathcal{X}} : T(\boldsymbol{x}) \in \mathrm{dom}\left(-\log_{\chi_{\tilde{Q}(\boldsymbol{x})}}\right)^{\star} \right\} \;. \tag{14}$$

Since $-\log_{\chi_{\tilde{Q}(\boldsymbol{x})}}$ is convex for any $\boldsymbol{x}$, it follows from Legendre duality,

$$KL_{\chi_{\tilde{Q}}}(\tilde{\mathbb{Q}}\|\mathbb{P}) \;=\; \mathbb{E}_{\mathsf{X}\sim\tilde{\mathbb{Q}}}\left[-\log_{\chi_{\tilde{Q}(\mathsf{X})}}\left(\frac{P(\mathsf{X})}{\tilde{Q}(\mathsf{X})}\right)\right]$$

$$=\; \mathbb{E}_{\mathsf{X}\sim\tilde{\mathbb{Q}}}\left[\sup_{T(\mathsf{X})\in\mathrm{dom}\left(\log_{\chi_{\tilde{Q}(\mathsf{X})}}\right)^{\star}}\left\{T(\mathsf{X})\cdot\frac{P(\mathsf{X})}{\tilde{Q}(\mathsf{X})} - (-\log_{\chi_{\tilde{Q}(\mathsf{X})}})^{\star}(T(\mathsf{X}))\right\}\right]$$

$$=\; \sup_{T\in\mathcal{F}_{\tilde{Q}}}\left\{\mathbb{E}_{\mathsf{X}\sim\tilde{\mathbb{Q}}}\left[T(\mathsf{X})\cdot\frac{P(\mathsf{X})}{\tilde{Q}(\mathsf{X})} - (-\log_{\chi_{\tilde{Q}(\mathsf{X})}})^{\star}(T(\mathsf{X}))\right]\right\}$$

$$=\; \sup_{T\in\mathcal{F}_{\tilde{Q}}}\left\{\mathbb{E}_{\mathsf{X}\sim\mathbb{P}}[T(\mathsf{X})] - \mathbb{E}_{\mathsf{X}\sim\tilde{\mathbb{Q}}}[(-\log_{\chi_{\tilde{Q}(\mathsf{X})}})^{\star}(T(\mathsf{X}))]\right\} \;. \tag{15}$$

Now, we know that $-\log_{\chi_{\tilde{Q}(\boldsymbol{x})}}(z)$ is proper lower-semicontinuous and therefore $(-\log_{\chi_{\tilde{Q}(\boldsymbol{x})}})^{\star\star} = -\log_{\chi_{\tilde{Q}(\boldsymbol{x})}}$. Being closed, the domain of the derivative of $(-\log_{\chi_{\tilde{Q}(\boldsymbol{x})}})^{\star}$ is the image of the derivative of $-\log_{\chi_{\tilde{Q}(\boldsymbol{x})}}$, given by $-\tilde{Q}(\boldsymbol{x})/\chi(\tilde{Q}(\boldsymbol{x})t)$. If $\chi : \mathbb{R}_+ \to \mathbb{R}_+$, then $-\tilde{Q}(\boldsymbol{x})/\chi(\tilde{Q}(\boldsymbol{x})t) \in \overline{\mathbb{R}_{++}}, \forall \tilde{Q}(\boldsymbol{x})$ and so $\mathcal{F}_{\tilde{Q}} = \left\{ T \in \overline{\mathbb{R}_{++}}^{\mathcal{X}} \right\}$.

A pointwise differentiation of eq. (15) yields that at the optimum, we have

$$
\begin{aligned}
P(\boldsymbol{x}) - \tilde{Q}(\boldsymbol{x}) \cdot (-\log_{\chi_{\tilde{Q}(\boldsymbol{x})}})^{\star\prime}(T(\boldsymbol{x})) &= P(\boldsymbol{x}) - \tilde{Q}(\boldsymbol{x}) \cdot (-\log_{\chi_{\tilde{Q}(\boldsymbol{x})}})^{\prime -1}(T(\boldsymbol{x})) \\
&= 0 \;,
\end{aligned}
\tag{16}
$$

that is, exploiting the fact that $(-\log_{\chi_{\tilde{Q}(\boldsymbol{x})}})' = -\tilde{Q}(\boldsymbol{x})/\chi(\tilde{Q}(\boldsymbol{x})t)$,

$$
\begin{aligned}
T^*(\boldsymbol{x}) &= (-\log_{\chi_{\tilde{Q}}})' \left( \frac{P(\boldsymbol{x})}{\tilde{Q}(\boldsymbol{x})} \right) \\
&= -\frac{\tilde{Q}(\boldsymbol{x})}{\chi\left( \frac{P(\boldsymbol{x})}{\tilde{Q}(\boldsymbol{x})} \cdot \tilde{Q}(\boldsymbol{x}) \right)} \\
&= -\frac{\tilde{Q}(\boldsymbol{x})}{\chi(P(\boldsymbol{x}))} \\
&= -\frac{1}{Z} \cdot \frac{\chi(Q(\boldsymbol{x}))}{\chi(P(\boldsymbol{x}))} \;.
\end{aligned}
\tag{17}
$$
$$
\tag{18}
$$

## IV   Proof of Theorem 9

In the context of the proof, we simplify notations and replace signature $\chi_{\text{net}}$ by $\chi$ and output activation $v_{\text{out}}$ by $v_2$. Let us call $\boldsymbol{z} \in \mathbb{R}^d$ the output of $g$. We revert the transformation and check:

$$
\begin{aligned}
\boldsymbol{\phi}_{l-1}(\boldsymbol{z}) &\doteq \mathrm{W}_l^{-1}(\boldsymbol{v}^{-1}(\boldsymbol{\phi}_l(\boldsymbol{z})) - \boldsymbol{b}_l) \;, \forall l \in \{1, 2, ..., L\} \;, \tag{19} \\
\boldsymbol{\phi}_L(\boldsymbol{z}) &= \Gamma^{-1}\left( \boldsymbol{v}_2^{-1}(\boldsymbol{z}) - \boldsymbol{\beta} \right) \;. \tag{20}
\end{aligned}
$$

For the sake of readability, we shall sometimes remove the dependence in $\boldsymbol{z}$. Letting $a_i$ denote coordinate $i$ in vector $\boldsymbol{a}$, $(\mathrm{A})_{ij}$ the coordinate in row $i$ and column $j$ of matrix A, for any $i, j \in [d]$, and $a_{l,i}$ coordinate $i$ in vector $\boldsymbol{a}_l$, we have

$$
\frac{\partial \phi_{l-1,i}}{\partial \phi_{l,j}} = (\mathrm{W}_l^{-1})_{ij} \cdot \frac{1}{v_i'(\boldsymbol{v}^{-1}(\boldsymbol{\phi}_l))} \;,
\tag{21}
$$

and furthermore

$$
\frac{\partial \phi_{L,i}}{\partial z_j} = (\Gamma^{-1})_{ij} \cdot \frac{1}{v_{2i}'(\boldsymbol{v}_2^{-1}(\boldsymbol{z}))} \;.
\tag{22}
$$

Let us denote vector $\tilde{\boldsymbol{a}}$ as the vector whose coordinates are the inverses of those of $\boldsymbol{a}$, namely $\tilde{a}_i \doteq 1/a_i$. From eqs. (21) and (22), the layerwise Jacobians are:

$$\frac{\partial \boldsymbol{\phi}_{l-1}}{\partial \boldsymbol{\phi}_l^\top} = \mathrm{W}_l^{-1} \odot \tilde{\boldsymbol{v}}'(\boldsymbol{v}^{-1}(\boldsymbol{\phi}_l))\mathbf{1}^\top \ , \forall l \in \{1, 2, ..., L\} \ , \tag{23}$$

$$\frac{\partial \boldsymbol{\phi}_L}{\partial \boldsymbol{z}^\top} = \Gamma^{-1} \odot \tilde{\boldsymbol{v}}_2'(\boldsymbol{v}_2^{-1}(\boldsymbol{z}))\mathbf{1}^\top \ , \tag{24}$$

where $\odot$ is Hadamard (coordinate-wise) product. These Jacobians have a very convenient form, since:

$$\begin{aligned}
\det\left(\frac{\partial \boldsymbol{\phi}_{l-1}}{\partial \boldsymbol{\phi}_l^\top}\right) &= \sum_{\boldsymbol{\sigma} \in S_d} \mathrm{sign}(\boldsymbol{\sigma}) \cdot \prod_{i=1}^d \left(\mathrm{W}_l^{-1} \odot \tilde{\boldsymbol{v}}'(\boldsymbol{v}^{-1}(\boldsymbol{\phi}_l))\mathbf{1}^\top\right)_{i,\sigma_i} \\
&= \sum_{\boldsymbol{\sigma} \in S_d} \mathrm{sign}(\boldsymbol{\sigma}) \cdot \prod_{i=1}^d (\mathrm{W}^{-1})_{l,i,\sigma_i} \left(\tilde{\boldsymbol{v}}'(\boldsymbol{v}^{-1}(\boldsymbol{\phi}_l))\mathbf{1}^\top\right)_{i,\sigma_i} \\
&= \sum_{\boldsymbol{\sigma} \in S_d} \left(\prod_{i=1}^d \tilde{v}'_i(\boldsymbol{v}^{-1}(\boldsymbol{\phi}_l))\right) \cdot \mathrm{sign}(\boldsymbol{\sigma}) \cdot \prod_{i=1}^d (\mathrm{W}^{-1})_{l,i,\sigma_i} \\
&= \left(\prod_{i=1}^d \tilde{v}'_i(\boldsymbol{v}^{-1}(\boldsymbol{\phi}_l))\right) \cdot \sum_{\boldsymbol{\sigma} \in S_d} \mathrm{sign}(\boldsymbol{\sigma}) \cdot \prod_{i=1}^d (\mathrm{W}^{-1})_{l,i,\sigma_i} \\
&= \left(\prod_{i=1}^d \tilde{v}'_i(\boldsymbol{v}^{-1}(\boldsymbol{\phi}_l))\right) \cdot \det\left(\mathrm{W}_l^{-1}\right) \\
&= \left(\prod_{i=1}^d \tilde{v}'_i(\boldsymbol{v}^{-1}(\boldsymbol{\phi}_l))\right) \cdot (\det\left(\mathrm{W}_l\right))^{-1} \ , \forall l \in \{1, 2, ..., L\} \ ,
\end{aligned}$$

and, using the same derivations,

$$\det\left(\frac{\partial \boldsymbol{\phi}_L}{\partial \boldsymbol{z}^\top}\right) = \left(\prod_{i=1}^d \tilde{v}'_{2i}(\boldsymbol{v}_2^{-1}(\boldsymbol{z}))\right) \cdot (\det\left(\Gamma\right))^{-1} \ . \tag{25}$$

The change of variable formula [4] yields:

$$
\begin{aligned}
Q_g(\boldsymbol{z}) &= Q_{\text{in}}(\boldsymbol{g}^{-1}(\boldsymbol{z})) \cdot \left| \det\left( \frac{\partial \boldsymbol{g}^{-1}}{\partial \boldsymbol{z}^\top} \right) \right| \\
&= Q_{\text{in}}(\boldsymbol{g}^{-1}(\boldsymbol{z})) \cdot \left| \det\left( \frac{\partial \boldsymbol{\phi}_0}{\partial \boldsymbol{z}^\top} \right) \right| \\
&= Q_{\text{in}}(\boldsymbol{g}^{-1}(\boldsymbol{z})) \cdot \left| \det\left( \prod_{l=1}^{L} \frac{\partial \boldsymbol{\phi}_{l-1}}{\partial \boldsymbol{\phi}_l^\top} \cdot \frac{\partial \boldsymbol{\phi}_L}{\partial \boldsymbol{z}^\top} \right) \right| \\
&= Q_{\text{in}}(\boldsymbol{g}^{-1}(\boldsymbol{z})) \cdot \left| \prod_{l=1}^{L} \det\left( \frac{\partial \boldsymbol{\phi}_{l-1}}{\partial \boldsymbol{\phi}_l^\top} \right) \cdot \det\left( \frac{\partial \boldsymbol{\phi}_L}{\partial \boldsymbol{z}^\top} \right) \right| \\
&= Q_{\text{in}}(\boldsymbol{g}^{-1}(\boldsymbol{z})) \cdot \prod_{l=1}^{L}\prod_{i=1}^{d} |\tilde{v}'_i(\boldsymbol{v}^{-1}(\boldsymbol{\phi}_l))| \cdot \prod_{i=1}^{d} |\tilde{v}'_{2i}(\boldsymbol{v}_2^{-1}(\boldsymbol{z}))| \cdot \left| \det\left( \Gamma \cdot \prod_{l=1}^{L} \mathrm{W}_l \right) \right|^{-1} \\
&= Q_{\text{in}}(\boldsymbol{g}^{-1}(\boldsymbol{z})) \cdot \frac{1}{\prod_{l=1}^{L}\prod_{i=1}^{d} |v'_i(\boldsymbol{v}^{-1}(\boldsymbol{\phi}_l))| \cdot \prod_{i=1}^{d} |v'_{2i}(\boldsymbol{v}_2^{-1}(\boldsymbol{z}))|} \cdot \left| \det\left( \Gamma \cdot \prod_{l=1}^{L} \mathrm{W}_l \right) \right|^{-1} \\
&= Q_{\text{in}}(\boldsymbol{g}^{-1}(\boldsymbol{z})) \cdot \frac{1}{\prod_{l=1}^{L}\prod_{i=1}^{d} |v'(v^{-1}(\phi_{l,i}))| \cdot \prod_{i=1}^{d} |v'_2(v_2^{-1}(z_i))|} \cdot \left| \det\left( \Gamma \cdot \prod_{l=1}^{L} \mathrm{W}_l \right) \right|^{-1} \\
&= \frac{Q_{\text{in}}(\boldsymbol{g}^{-1}(\boldsymbol{z}))}{\prod_{l=1}^{L}\prod_{i=1}^{d} |v'(v^{-1}(\phi_{l,i}))|} \cdot \frac{1}{\prod_{i=1}^{d} |v'_2(v_2^{-1}(z_i))| \cdot |\det(\mathrm{N})|} ,
\end{aligned}
$$

because $v$ and $v_2$ are coordinatewise. We have let

$$
\mathrm{N} = \Gamma \cdot \prod_{l=1}^{L} \mathrm{W}_l , \tag{26}
$$

and also $\phi_{l,i} \doteq v(\boldsymbol{w}_{l,i}^\top \boldsymbol{\phi}_{l-1} + b_{l,i})$, where $\boldsymbol{w}_{l,i} \doteq \mathrm{W}_l^\top \mathbf{1}_i$ is the (column) vector built from row $i$ in $\mathrm{W}_l$ and similarly $z_i \doteq v_2(\boldsymbol{\gamma}_i^\top \boldsymbol{\phi}_L + \beta_i)$ with $\boldsymbol{\gamma}_i \doteq \Gamma^\top \mathbf{1}_i$. Notice that we can also write

$$
\prod_{l=1}^{L}\prod_{i=1}^{d} |v'(v^{-1}(\phi_{l,i}))| = \prod_{l=1}^{L}\prod_{i=1}^{d} |v'(\boldsymbol{w}_{l,i}^\top \boldsymbol{\phi}_{l-1} + b_{l,i})| . \tag{27}
$$

So, letting $\tilde{Q}_{\text{deep}}^* \doteq \prod_{l=1}^{L}\prod_{i=1}^{d} |v'(\boldsymbol{w}_{l,i}^\top \boldsymbol{\phi}_{l-1} + b_{l,i})|$, $H_{\text{out}} \doteq \prod_{i=1}^{d} |v'_{\text{OUT}}(\boldsymbol{\gamma}_i^\top \boldsymbol{\phi}_L(\boldsymbol{x}) + \beta_i)|$ (with $\boldsymbol{x} \doteq \boldsymbol{g}^{-1}(\boldsymbol{z})$), and dropping the determinant which does not depend on $\boldsymbol{z}$, we get:

$$
Q_g(\boldsymbol{z}) \propto \frac{Q_{\text{in}}(\boldsymbol{g}^{-1}(\boldsymbol{z}))}{\tilde{Q}_{\text{deep}}^*} \cdot \frac{1}{H_{\text{out}}} . \tag{28}
$$

To finish up the proof, we are going to identify $\tilde{Q}_{\text{deep}}^*$ to (a constant times) the product of escorts in eq. (14). To do so, we are first going to design the general activation function $v$ as a function of $\chi$, and

choose:

$$v(z) \;\doteq\; k + k' \cdot \exp_\chi(z) \;,\tag{29}$$

for $k \in \mathbb{R}, k' > 0$ constants, which can be chosen *e.g.* to ensure that zero signal implies zero activation ($v(0) = 0$). Our choice for $v$ has the following key properties.

**Lemma B** *$v$ is $C^1$, invertible and we have $v'(z) = k' \cdot \chi(\exp_\chi(z))$.*

**Proof** The derivative comes from [2, Eq. 84]. Notice that $\exp_\chi$ is continuous as an integral, $\chi$ is continuous by assumption and so $v'$ is continuous, implying $v$ is $C^1$. We prove the invertibility. Because of the expression of $v'$, $v$ is increasing, and in fact strictly increasing with the sole exception when $\exp_\chi(z) \in \chi^{-1}(0)$. Hovever, note that $\chi^{-1}(0) \not\subset \mathrm{dom}(\log_\chi)$ because of the definition of $\log_\chi$. Since $\exp_\chi$ is the inverse of $\log_\chi$ [9, Section 10.1], it follows that $\chi^{-1}(0) \not\subset \mathrm{im}(\exp_\chi)$ and so $\exp_\chi(z) \notin \chi^{-1}(0), \forall z \in \mathrm{dom}(\exp_\chi)$, which implies $v$ invertible. ∎

What the Lemma shows is that we can plug $v$ as in eq. (29) directly in $\tilde{Q}^*_{\mathrm{deep}}$. To do so, let us now define strictly positive constants $Z_{li}$ that shall be fixed later. We directly get from eq. (27)

$$\begin{aligned}
\tilde{Q}^*_{\mathrm{deep}} &= \left( \prod_{l=1}^L \prod_{i=1}^d Z_{li} \right) \cdot \prod_{l=1}^L \prod_{i=1}^d \frac{1}{Z_{li}} \cdot |v'(\boldsymbol{w}_{l,i}^\top \boldsymbol{\phi}_{l-1} + b_{l,i})| \\
&= \left( k'^{Ld} \cdot \prod_{l=1}^L \prod_{i=1}^d Z_{li} \right) \cdot \underbrace{\prod_{l=1}^L \prod_{i=1}^d \frac{1}{Z_{li}} \cdot \chi(\exp_\chi(\boldsymbol{w}_{l,i}^\top \boldsymbol{\phi}_{l-1} + b_{l,i}))}_{\doteq \tilde{Q}_{\mathrm{deep}}}
\end{aligned}\tag{30}$$

(we can remove the absolute values since $\chi$ is non-negative). We now ensure that $\tilde{Q}_{\mathrm{deep}}$ is indeed a product of escorts: to do so, we just need to ensure that (i) $b_{l,i}$ normalizes the deformed exponential family, *i.e.* defines (negative) its cumulant (Definition 2), and (ii) $Z_{li}$ normalizes its escort as in eq. (6). To be more explicit, we pick $b_{l,i}$ the solution of

$$\int_{\boldsymbol{\phi}} \exp_\chi(\boldsymbol{w}_{l,i}^\top \boldsymbol{\phi} - b_{l,i}) \mathrm{d}\nu_{l-1}(\boldsymbol{\phi}) \;=\; 1 \;,\tag{31}$$

where $\mathrm{d}\nu_{l-1}(\boldsymbol{\phi}) \doteq \int_{\boldsymbol{\phi}_{l-1}(\boldsymbol{x})=\boldsymbol{\phi}} \mathrm{d}\mu(\boldsymbol{x})$ is the pushforward measure, and

$$Z_{li} \;=\; \int_{\boldsymbol{x}} \chi(P_{\chi,b_{l,i}}(\boldsymbol{x}|\boldsymbol{w}_{l,i}, \boldsymbol{\phi}_{l-1})) \mathrm{d}\mu(\boldsymbol{x}) \;.\tag{32}$$

We get

$$\tilde{Q}_{\mathrm{deep}} = \; \prod_{l=1}^L \prod_{i=1}^d \tilde{P}_{\chi,b_{l,i}}(\boldsymbol{x}|\boldsymbol{w}_{l,i}, \boldsymbol{\phi}_{l-1}) \;,\tag{33}$$

and finally,

$$\begin{aligned}
Q_g(\boldsymbol{z}) &= \frac{Q_{\mathrm{in}}(\boldsymbol{x})}{\tilde{Q}^*_{\mathrm{deep}}(\boldsymbol{x})} \cdot \frac{1}{H_{\mathrm{out}}(\boldsymbol{x}) \cdot |\det(\mathbf{N})|} \\
&= \frac{Q_{\mathrm{in}}(\boldsymbol{x})}{\tilde{Q}_{\mathrm{deep}}(\boldsymbol{x})} \cdot \frac{1}{H_{\mathrm{out}}(\boldsymbol{x}) \cdot Z_{\mathrm{net}}} \;,
\end{aligned}\tag{34}$$

with

$$Z_{\text{net}} \quad \dot{=} \quad \left( k'^{Ld} \cdot \prod_{l=1}^{L} \prod_{i=1}^{d} Z_{li} \right) \cdot |\det(\mathbf{N})| \tag{35}$$

a constant. We get the statement of Theorem 9.

**Remark.** (unnormalized densities) since in practice all $\boldsymbol{b}_l$s are learned, we in fact work with deformed exponential families with unspecified normalization. We may also consider that the normalization of escorts is unspecified and therefore drop all $Z_{li}$s, which simplifies $Z_{\text{net}}$ to $Z_{\text{net}} = |\det(\mathbf{N})|$. ■

**Remark.** (completely factoring $Q_g$ as an escort) Denote for short $\boldsymbol{z}_p \dot{=} \boldsymbol{\phi}_L(\boldsymbol{x})$ the penultimate layer of $\boldsymbol{g}$, and $\boldsymbol{g}_p$ the net obtain from eliminating the last layer of $\boldsymbol{g}$, which allows us to drop $H_{\text{out}}(.)$ from $Q_{g_p}(\boldsymbol{z})$ and we have $Q_g(\boldsymbol{z}) \propto Q_r \dot{=} Q_{\text{in}}(\boldsymbol{g}_p^{-1}(\boldsymbol{z}_p))/\tilde{Q}_{\text{deep}}(\boldsymbol{g}_p^{-1}(\boldsymbol{z}_p))$. One can factor $Q_r$ as a proper likelihood over escorts of $\chi_{\text{net}}$-exponential families: for this, replace all $Ld$ inner nodes of $\boldsymbol{g}_p$ by random variables, say $\Phi_{l,i}$ (for $l \in \{0, 1, ..., L{-}1\}, i \in \{1, 2, ..., d\}$), treat the deep net $\boldsymbol{g}_p$ as a directed graphical model whose connections are the dashed arcs. Now, if we let, say, $Q_{\text{in}}(\boldsymbol{g}_p^{-1}(\boldsymbol{z}_p)) \dot{=} \tilde{Q}_a(\cap_{l,i}\Phi_{l,i})$ and $\tilde{Q}_{\text{deep}}(\boldsymbol{g}_p^{-1}(\boldsymbol{z}_p)) \dot{=} \tilde{Q}_b(\cap_{l>0,i}\Phi_{l,i})$, and if we use as $Q_{\text{in}}$ an uninformed escort (*i.e.* with constant coordinate, say for example $\boldsymbol{\theta} = \mathbf{1}$, Definition 2), then assuming correct factorization one may obtain $Q_r = \tilde{Q}_c(\boldsymbol{g}_p^{-1}(\boldsymbol{z}_p)| \cap_{l>0,i} \Phi_{l,i})$ for some escort $\tilde{Q}_c$ that we can plug directly in eq. (11). To properly understand the relationships between $\chi, Q_a, Q_b$ and how the escorts factor in $Q_c$ requires a push of the state of the art: conjugacy in deformed exponential families is less understood than for exponential families; it is also unknown how product of deformed exponential families factor within the same deformed exponential families [1]; some factorizations are known but only on subsets of deformed exponential families and rely on particular notions of independence [8]; ■

**Remark.** (twist introduced by the last layer) We return to the twist introduced by the last layer of $\boldsymbol{g}$:

$$H_{\text{out}}(\boldsymbol{x}) \quad = \quad \prod_{i=1}^{d} |v_2'(\boldsymbol{\gamma}_i^\top \boldsymbol{\phi}_L(\boldsymbol{x}) + \beta_L)| \ . \tag{36}$$

It is clear that when $v_2$ is the identity, $H_{\text{out}}(\boldsymbol{x})$ is constant; so deep architectures, as experimentally carried out *e.g.* in Wasserstein GANs [3] or analyzed theoretically *e.g.* in [7] exactly fit to the escort factoring — notice that one can choose as input density one from some particular deformed exponential family, as *e.g.* done experimentally for [11, Section 2.5] (standard Gaussian), so that in this case $Q_g(\boldsymbol{z})$ factors completely as escorts.

Suppose now that $v_2$ is not the identity but chosen so that, for some couple $(\chi, g)$ where $\chi$ is differentiable and $g : \mathbb{R}_+ \to \mathbb{R}$ is invertible,

$$(v_2' \circ g)(z) \quad = \quad \frac{\mathrm{d}}{\mathrm{d}z}(\log_\chi \circ \chi)(z) = \frac{\chi'(z)}{\chi(z)} \ , \tag{37}$$

which is equivalent, after a variable change, to having $v_2$ satisfy

$$v_2'(t) \quad = \quad \frac{\chi' \circ g^{-1}}{\chi \circ g^{-1}}(t) \ . \tag{38}$$

In addition, suppose that $g$ is chosen so that $\sum_i g^{-1}(\boldsymbol{\gamma}_i^\top \boldsymbol{\phi}_L(\boldsymbol{x}) + \beta_i) = 1$. Call $D \doteq \{p_1, p_2, ..., p_d\}$ this discrete distribution, removing reference to $\boldsymbol{x}$. We then have:

$$
\begin{aligned}
H_{\text{out}}(\boldsymbol{x}) &= \prod_{i=1}^{d} \frac{\chi'(g^{-1}(\boldsymbol{\gamma}_i^\top \boldsymbol{\phi}_L(\boldsymbol{x}) + \beta_i))}{\chi(g^{-1}(\boldsymbol{\gamma}_i^\top \boldsymbol{\phi}_L(\boldsymbol{x}) + \beta_i))} \\
&= \prod_{i=1}^{d} \frac{\chi'(p_i)}{\chi(p_i)} \\
&= \prod_{i=1}^{d} \chi(p_i) \cdot \frac{\chi'(p_i)}{\chi(p_i)} \\
&= \left| \prod_{i=1}^{d} ((\exp_\chi)' \circ \log_\chi)(p_i) \cdot (\log_\chi)''(p_i) \right| \\
&\propto |\det(H)| \ .
\end{aligned}
\tag{39}
$$

Here, $H$ is the $\chi$-Fisher information metric of $D$ [2, Theorem 12, eqs 119, 120]. In other words, $H_{\text{out}}(\boldsymbol{x})$ can be absorbed in the volume element in eq. (34).

As an example, pick a prop-$\tau$ activation (Table 1), for which $\log_\chi = (\tau^\star)^{-1}(\tau^\star(0)z)$ and

$$
\chi(t) = \frac{(\tau^\star)' \circ (\tau^\star)^{-1}(\tau^\star(0)z)}{\tau^\star(0)} \ .
\tag{40}
$$

Now, pick $g(z) = \log_\chi(K \cdot z)$, where $K \doteq \sum_i \exp_\chi(\boldsymbol{\gamma}_i^\top \boldsymbol{\phi}_L(\boldsymbol{x}) + \beta_i)$ guarantees:

$$
\sum_i g^{-1}(\boldsymbol{\gamma}_i^\top \boldsymbol{\phi}_L(\boldsymbol{x}) + \beta_i) = \frac{1}{K} \cdot \sum_i \exp_\chi(\boldsymbol{\gamma}_i^\top \boldsymbol{\phi}_L(\boldsymbol{x}) + \beta_i) = 1 \ .
\tag{41}
$$

Condition in eq. (37) becomes

$$
\begin{aligned}
(v_2' \circ (\tau^\star)^{-1})(\tau^\star(0)Kz) &= \frac{\chi' \circ g^{-1}}{\chi \circ g^{-1}}(t) \\
&= \tau^\star(0) \cdot \frac{(\tau^\star)'' \circ (\tau^\star)^{-1}(\tau^\star(0)Kz)}{((\tau^\star)' \circ (\tau^\star)^{-1}(\tau^\star(0)Kz))^2} \ ,
\end{aligned}
\tag{42}
$$

and we obtain after a variable change,

$$
v_2 = \tau^\star(0) \cdot \int_t \frac{(\tau^\star)''(t)}{((\tau^\star)')^2(t)} \mathrm{d}t \ ,
\tag{43}
$$

which does not depend on $K$ and, if $\tau^\star$ is *strictly* convex, is strictly increasing. Notice that we can carry out the integration, $v_2(z) = K' - (\tau^\star(0)/(\tau^\star)'(z))$ for some constant $K'$. To make a parallel with a popular activation for the last layer, consider the sigmoid, $v_2 \doteq v_s(z) \doteq 1/(1 + \exp(-z))$, for which

$$
v_s'(z) = \frac{\exp(z)}{(1 + \exp(z))^2} \ .
\tag{44}
$$

Fitting it to eq. (43),

$$\frac{\exp(z)}{(1 + \exp(z))^2} \quad = \quad \tau_s^\star(0) \cdot \frac{(\tau_s^\star)''(t)}{((\tau_s^\star)')^2(t)} \tag{45}$$

reveals that we can pick $\tau_s^\star(z) = z + \exp(z)$ (we control that $\tau_s^\star(0) = 1$). Such a $\tau^\star$ analytically fits to the prop-$\tau$ definition and in fact corresponds to a $\chi$-exponential family, but it does not correspond to an entropy $\tau$. This would be also true for affine scalings (argument and function) of the sigmoid of the type $v_2 = a + bv_s(c + dz)$. ∎

# V  Proof of Lemma 11

We first show point (i). Define function

$$h(z) \quad \dot{=} \quad \frac{v(z) - \inf v(z)}{v(0) - \inf v(z)} \quad, \tag{46}$$

and let $g(z) \doteq h^{-1}(z)$. Since $\mathrm{dom}(v) \cap \overline{\mathbb{R}_+} \neq \emptyset$, $v(0) - \inf v(z) > 0$, so $h(z)$ bears the same properties as $v$. We first show that $g$ is a valid $\chi$-logarithm. Since $v$ is convex increasing, $g(z)$ is concave increasing and $-g$ is convex decreasing. Therefore, since $g$ is $C^1$ as well, letting $\xi \doteq g'$, we get:

$$g(z) \quad = \quad \int_1^z \frac{1}{\left(\frac{1}{\xi(t)}\right)} \mathrm{d}t \quad . \tag{47}$$

We also check that $g(1) = 0$ since $h(0) = 1$. If we let $\chi \doteq 1/\xi$, then because $\xi(z) \geq 0$, $\chi(z) \geq 0$ and also because $\xi$ is decreasing, $\chi$ is increasing. Finally, $\chi : \mathbb{R}_+ \to \mathbb{R}_+$. Summarizing, we have shown that $\chi$ defines a valid signature and $g(z) = \log_\chi(z)$. Therefore, $h(z) = \exp_\chi(z)$ and it comes that

$$v(z) \quad = \quad k + k' \cdot \exp_\chi(z) \quad , \tag{48}$$

for $k \doteq \inf v(z) \in \mathbb{R}$ and $k' \doteq v(0) - \inf v(z) > 0$, so $v$ matches the analytic expression in eq. (29), which allows to complete the proof of point (i) in the Lemma.

The strong admissibility results are easy to check in point (ii), so we concentrate on showing the weak admissibility of ReLU. We use a scaled perspective transform of the Softplus function and let:

$$v_\mu(z) \quad \dot{=} \quad (1 - \mu) \cdot \log\left(1 + \exp\left(\frac{z}{1 - \mu}\right)\right) \quad , \tag{49}$$

with $\mu \in [0, 1]$. It is clear that $v_\mu$ is strongly admissible for any $\mu \in [0, 1)$.

**Lemma C** *For any $z \geq 0, \mu \in [0, 1]$,*

$$(1 - \mu) \cdot \log\left(\frac{1 + \exp\left(\frac{z}{1-\mu}\right)}{1 + \exp(z)}\right) \quad \leq \quad \mu z \quad . \tag{50}$$

**Proof** Equivalently, we want

$$\frac{1 + \exp\left(\frac{z}{1-\mu}\right)}{1 + \exp(z)} \leq \exp\left(\frac{\mu z}{1-\mu}\right) \ , \tag{51}$$

or, equivalently,

$$1 + \exp\left(\frac{z}{1-\mu}\right) \leq \exp\left(\frac{\mu z}{1-\mu}\right) + \exp(z) \cdot \exp\left(\frac{\mu z}{1-\mu}\right)$$

$$= \exp\left(\frac{\mu z}{1-\mu}\right) + \exp\left(\frac{z}{1-\mu}\right) \ , \tag{52}$$

which, after simplification, is equivalent to $\mu z / (1-\mu) \geq 0$, which indeed holds when $z \geq 0, \mu \in [0,1]$. ∎

We now have $v_\mu(z) \geq \max\{0, z\}, \forall \mu \in [0,1]$, and we can also check that Lemma C implies

$$(1-\mu) \cdot \log\left(1 + \exp\left(\frac{z}{1-\mu}\right)\right) - z$$

$$\leq (1-\mu) \cdot (\log(1 + \exp(z)) - z) \ , \forall z \geq 0, \mu \in [0,1] \ . \tag{53}$$

Let us denote, for any $z \geq 0$,

$$I_\mu(z) \doteq \int_0^z |v_\mu(t) - \max\{0, t\}| \mathrm{d}t$$

$$= \int_0^z |v_\mu(t) - t| \mathrm{d}t$$

$$= \int_0^z (v_\mu(t) - t) \mathrm{d}t \ . \tag{54}$$

Since $\max\{0, -t\} = \max\{0, t\} - t$ and $v_\mu(-t) = v_\mu(t) - t$, we have $\|v_\mu - \mathrm{ReLU}\|_{L1} = 2 \lim_{z \to +\infty} I_\mu(z)$. It also comes from ineq. (53) that

$$I_\mu(z) \leq (1-\mu) I_0(z) \ , \forall z \leq 0 \ , \tag{55}$$

furthermore, it can be shown by numerical integration that $\lim_{+\infty} I_0(z) = \pi^2 / 6$, so we get

$$\|v_\mu - \mathrm{ReLU}\|_{L1} \leq \frac{(1-\mu)\pi^2}{3} \ , \forall \mu \in [0,1] \ , \tag{56}$$

and to have the right hand side smaller than $\epsilon > 0$, it suffices to take

$$\mu > 1 - \frac{3\epsilon}{\pi^2} \ , \tag{57}$$

which completes the proof of point (ii) and so the statement of the Lemma.

# VI    Proof of Theorem 12

The proof of the Theorem mainly follows from identifying the parameters of eq. (17) with the variational part of eq. (11). Recall that

$$(\chi^\bullet)_{\frac{1}{q}}(t) = \frac{1}{(\chi_q)^{-1}\left(\frac{1}{t}\right)} ,\tag{58}$$

so, exploiting the fact that $K(\mathbb{Q})$ does not depend on $T$, we get:

$$
\begin{aligned}
\ell'_{\boldsymbol{x}}(-1, z) &= \frac{\mathrm{d}}{\mathrm{d}z}(-\log_{\chi_{\tilde{Q}(\boldsymbol{x})}})^\star(-z) \\
&= \frac{\mathrm{d}}{\mathrm{d}z} - \log_{(\chi^\bullet)_{\frac{1}{\tilde{Q}(\boldsymbol{x})}}}(-z) \\
&= \left(\chi_{\tilde{Q}(\boldsymbol{x})}\right)^{-1}\left(-\frac{1}{z}\right) .
\end{aligned}\tag{59}
$$

Since $\ell'(+1, z) = -1$, we deduce that the loss is proper composite with inverse link function [13, Corollary 12] given by:

$$
\begin{aligned}
\Psi_{\boldsymbol{x}}^{-1}(z) &= \frac{\ell'(-1, z)}{\ell'(-1, z) - \ell'(+1, z)} \\
&= \frac{\left(\chi_{\tilde{Q}(\boldsymbol{x})}\right)^{-1}\left(-\frac{1}{z}\right)}{\left(\chi_{\tilde{Q}(\boldsymbol{x})}\right)^{-1}\left(-\frac{1}{z}\right) + 1} ,
\end{aligned}\tag{60}
$$

so that the link is

$$\Psi_{\boldsymbol{x}}(z) = -\frac{1}{\chi_{\tilde{Q}(\boldsymbol{x})}\left(\frac{z}{1-z}\right)} .\tag{61}$$

**Remark.**    We easily retrieve the optimal discriminator (Theorem 7) but this time from the proper composite loss, since (the first line is a general property of $\Psi_{\boldsymbol{x}}$, see Section 5):

$$
\begin{aligned}
T^*(\boldsymbol{x}) &= \Psi_{\boldsymbol{x}}\left(\frac{P(\boldsymbol{x})}{P(\boldsymbol{x}) + \tilde{Q}(\boldsymbol{x})}\right) \\
&= -\frac{1}{\chi_{\tilde{Q}(\boldsymbol{x})}\left(\frac{\frac{P(\boldsymbol{x})}{P(\boldsymbol{x})+\tilde{Q}(\boldsymbol{x})}}{1-\frac{P(\boldsymbol{x})}{P(\boldsymbol{x})+\tilde{Q}(\boldsymbol{x})}}\right)} . \\
&= -\frac{1}{\chi_{\tilde{Q}(\boldsymbol{x})}\left(\frac{P(\boldsymbol{x})}{\tilde{Q}(\boldsymbol{x})}\right)} \\
&= -\frac{1}{Z} \cdot \frac{\chi(Q(\boldsymbol{x}))}{\chi(P(\boldsymbol{x}))} .
\end{aligned}
$$

The last identity follows from eqs. (17) — (18).    ∎

# Supplementary material on experiments

# VII Architectures

We consider two architectures in our experiments: DCGAN [12] and the multilayer feedforward network (MLP) used in [11]. Suppose the size of input images is isize-by-isize, the details of architectures are given as follows:

**Generator of DCGAN** :
ConvTranspose(input=100, output=8×isize, stride=1) → BatchNorm→ Activation→ Conv(input=8×isize, output=4×isize, stride=2, padding=1)→ BatchNorm→ Activation→ ConvTranspose(input=4×isize, output=2×isize, stride=2, padding=2)→ BatchNorm→ Activation→ ConvTranspose(input=2×isize, output= isize, stride=2, padding=1)→ BatchNorm→ Activation → Conv(isize, number of channel, stride=2, padding=1) → Last Activation

**Discriminator of DCGAN** :
Conv(1, 2×isize, stride=2) → BatchNorm→ LeakyReLU→ Conv(input=2×isize, output=4×isize, stride=2, padding=1)→ BatchNorm→ LeakyReLU→ Conv(input=4×isize, output=8×isize, stride=2, padding=2)→ BatchNorm→ LeakyReLU→ Conv(input=8×isize, output= 1, stride=2, padding=1)→ Link function

**Generator of MLP** :
$z$ → Linear(100, 1024) → BatchNorm → Activation → Linear(1024, 1024) → BatchNorm → Activation → Linear(1024, isize×isize) → last Activation

**Discriminator of MLP** :
$x$ → Linear(isize×isize, 1024) → ELU → Linear(1024, 1024) → ELU → Linear(1024, 1) → Link function

# VIII Experimental setup for varying the activation function in the generator

**Setup.** We train adversarial networks with varying activation functions for the generators on the MNIST [6] and LSUN [15] datasets. In particular, we compare ReLU, Softplus, Least Square loss as an example of prop-$\tau$, and $\mu$-ReLU with varying $\mu$ in $[0, 0.1, ..., 1]$ by using them as the activation functions in all hidden layers of the generators. For all models, we fix the learning rate to 0.0002 and batch size to 64 throughout all experiments after tuning on a hold-out set.

**MNIST.** We evaluate the activation functions by using both DCGAN and the MLP used in [11] as the architectures. As training divergence, we adopt both GAN and Wasserstein distance (WGAN) because GAN belongs to variational $f$-divergence formulation while WGAN does not. The link function of the discriminators is specific to the respective divergence, which is sigmoid for GAN and linear for WGAN. We sample random noise $z \in \text{Uniform}_{100}(0, 1)$ for MLP and $z \in \text{Gaussian}(0, 1)$ for DCGAN, which is found slightly better than sampling from $\text{Uniform}_{100}(-1, 1)$. As the best practice, we apply Adam [5] to optimize models with GAN and RMSprop [14] to optimize WGAN based models. For GAN, we train one batch for discriminator and one batch for generator iteratively during training. For WGAN, we apply weight clipping with 0.01 and train five batches for discriminator and one batch for generator interchangeably during training.

We train all models on the full MNIST training data set and evaluate the performance on the test set by using the kernel density estimation (KDE). Since the size of images accepted by DCGAN should be n-fold of 16, all images are rescaled to 32-by-32 for all models. Following [11], we apply three-fold cross validation to find optimal bandwidth for the isotropic Gaussian kernel of KDE on a hold-out set. To estimate the log probability of the test set, we sample 16k images from the models in the same way as [11]. We observe that the initialization of model parameters has significant influence on performance. Therefore, we conduct three runs with different random seeds for each experimental setting and report the mean and standard deviation of the results.

**LSUN.** We also evaluate all activation functions in consideration for the generator on LSUN natural scene images. We train DCGAN with GAN as the divergence on the *tower* category of images, which are rescaled and center-cropped to 64-by-64 pixels, as in [12]. Due to the center-cropped images, we apply `tanh` as last activation of generators instead of sigmoid for GAN based models.

# IX   Visual results on MNIST

| $\mu$ | | $\mu$ | |
|---|---|---|---|
| 0 |  | 0.1 |  |
| 0.2 |  | 0.3 |  |
| 0.4 |  | 0.5 |  |
| 0.6 |  | 0.7 |  |
| 0.8 |  | 0.9 |  |
| 1 |  | | |

Table A1: MNIST results for GAN_DCGAN at varying $\mu$ ($\mu = 1$ is ReLU).

| $\mu$ | | $\mu$ | |
|---|---|---|---|

Table A2: MNIST results for WGAN_DCGAN at varying $\mu$ ($\mu = 1$ is ReLU).

| $\mu$ | | $\mu$ | |
|---|---|---|---|
| 0 |  | 0.1 |  |
| 0.2 |  | 0.3 |  |
| 0.4 |  | 0.5 |  |
| 0.6 |  | 0.7 |  |
| 0.8 |  | 0.9 |  |
| 1 |  | | |

Table A3: MNIST results for WGAN_MLP at varying $\mu$ ($\mu = 1$ is ReLU).

| $\mu$ | | $\mu$ | |
|---|---|---|---|
| 0 |  | 0.1 |  |
| 0.2 |  | 0.3 |  |
| 0.4 |  | 0.5 |  |
| 0.6 |  | 0.7 |  |
| 0.8 |  | 0.9 |  |
| 1 |  | | |

Table A4: MNIST results for GAN_MLP at varying $\mu$ ($\mu = 1$ is ReLU).

| $\mu$ | | $\mu$ | |
|---|---|---|---|
| 0 |  | 0.1 |  |
| 0.2 |  | 0.3 |  |
| 0.4 |  | 0.5 |  |
| 0.6 |  | 0.7 |  |
| 0.8 |  | 0.9 |  |
| 1 |  | | |

Table A5: LSUN results for GAN_DCGAN at varying $\mu$ ($\mu = 1$ is ReLU).