[Reviews · NeurIPS 2017]

Reviewer 1



This paper considers the f-GAN principle generalized to f-divergences. Generalizing the connection between the KL divergence and regular exponential families by using the chi-logarithm, the authors prove the variational information geometric f-GAN (vig-f-GAN) identity, which provides an interpretation of the objective function of the f-GAN. The authors also prove a sufficient condition on activation functions to obtain factored generative distributions. The main theoretical result, vig-f-GAN identity is interesting and its applicability is promising. However, the meanings of main theorems are hard to follow since they are written in quite general forms. Section 6: It is hard to understand how the observations in experiments are related to theoretical results. p.7, l.258, For eq.(7), is means when C …’’: This clause has a wrong structure.

Reviewer 2



Thank you for an interesting read. This paper proposed an information geometric (IG) view of the f-GAN algorithm. It first showed that f-GAN converges in parameter space using the 1-1 mapping of f-divergence and chi-divergence and a Bregman divergence result. Then it also discussed a proper way to implement f-GAN. Finally in the main text it provided a factorisation result of the deep neural network representation and discussed the choice of activation functions which has 1-1 mapping to the chi (or f) function. I didn't check every detail in the appendix but it seems to me that the proofs (except for Thm. 8 which I don't have time to read before due) are correct. I think this paper is very dense and contains many new results that could be of interest for machine learning and information theory. Especially I'm impressed to see the exposition of Thm 4 which tells the IG part of the story for f-GAN. So I think this paper is a clear accept. However, I do think this paper could be very difficult to understand for those who don't know too much about the connections between IT and IG (at least the simplest one that KL for an exponential family can be mapped to Bregman divergence, and the Fenchel duality stuff). Many deep learning engineers could be this kind of person and also they're the main audience for a GAN paper. So the following are my suggestions that could potentially make the paper clearer: 1. I like Fig 1 in the appendix which states the connections between IT and IG in f-GAN context. Might consider moving it to the main text. 2. Might be helpful to make a clear statement why considering the IG view could be helpful at the beginning. My understanding is that you can use the geometry in the parameter space to discuss the behaviour of optimisation. You mentioned that briefly in page 7 which I found is a bit too late. 3. I'm not sure if I understand how section 5 connects to f-GAN. It seems to me that you just used the deformed exponential family to explain the distribution a deep neural net can represent, thus not a consequence of f-GAN optimisation results. Yes I buy the point that v, f, and chi have 1-1 correspondence, but then you didn't say anything about how this result could help design the f-GAN game, e.g. which f divergence we should pick, or given an activation function, which f-GAN objective works the best in terms of say convergence. 4. Why for Thm 6 the phi_l can be viewed as "deep sufficient statistics"? I don't think eq. (13) is of the form of a deformed exponential family? 5. As said, might be helpful to consider moving line 257-269 to other places. Also it seems to me that the utility theory part is not directly related to the IG view, so might be good to delete that paragraph (you can keep it in the appendix) and free some spaces to explain your main results. 6. I feel the experiments are not directly related to the main points claimed in the paper. For example, you can discuss (A) by only having the results from section 5, i.e. I don't really need to know the IG view of f-GAN to apply these new activation functions. Also for (B) I only need to understand section 4, which is not that closely related to the IG view of f-GAN. Especially your results of WGAN could be distractive and confusing, since this paper is mainly about f-GANs, and I actually spent some time to find the sentence (line 227-228) about WGAN. In summary, while this paper provides many useful results and dense derivations, I have a feeling that the material is not organised in a crystal clear way. Instead, it looks like squeezing results from multiple papers to an 8-page NIPS submission. So while I am supportive for acceptance, I do think this paper needs editing to make the claims clearer and more coherent.

Reviewer 3



The authors identify several interesting GAN-related questions, such as to what extend solving the GAN problem implies convergence in parameter space, what the generator is actually fitting when convergence occurs and (perhaps most relevant from a network architectural point of view) how to choose the output activation function of the discriminator so as to ensure proper compositeness of the loss function. The authors set out to address these questions within the (information theoretic) framework of deformed exponential distributions, from which they derive among other things the following theoretical results: They present a variational generalization (amenable to f-GAN formulation) of a known theorem that relates an f-divergence between distributions to a corresponding Bregman divergence between the parameters of such distributions. As such, this theorem provides an interesting connection between the information-theoretic view point of measuring dissimilarity between probability distributions and the information-geometric perspective of measuring dissimilarity between the corresponding parameters. They show that under a reversibility assumption, deep generative networks factor as so called escorts of deformed exponential distributions. Their theoretical investigation furthermore suggests that a careful choice of the hidden activation functions of the generator as well as a proper selection of the output activation function of the discriminator could potentially help to further improve GANs. They also briefly mention an alternative interpretation of the GAN game in the context of expected utility theory. Overall, I find that the authors present some interesting theoretical results, however, I do have several concerns regarding the practical relevance and usefulness of their results in the present form. Questions & concerns: To begin with, many of their theorems and resulting insights hold for the specific case that the data and model distributions P and Q are members of the deformed exponential family. Can the authors justify this assumption, i.e. elaborate on whether it is met in practice and explain whether similar results (such as Theorems 4 & 6) could be derived without this assumption? One of the author’s main contributions, the information-geometric f-GAN identity in Eq.(7), relates the well-known variational f-divergence formulation over distributions to an information-geometric optimization problem over parameters. Can the authors explain what we gain from this parameter-based point of view? Is it possible to implement GANs in terms of this parameter-based optimization and what would be the benefits? I would really have liked to see experimental results comparing the two approaches to optimization of GANs. Somewhat surprising, the authors don’t seem to make use of the right hand side of this identity, other than to assert that solving the GAN game implies convergence in parameter space (provided the residual J(Q) is small). And why is this implication not obvious? Can the authors give a realistic scenario where convergence in the variational f-divergence formulation over distributions does not imply convergence in parameter space? As an interesting practical implication of their theoretical investigation, the authors show that the choice of the output activation function g_f (see section 2.4 in Ref.[34]) matters in order for the GAN loss to have the desirable proper compositeness property. I found this result particularly interesting and would certainly have liked to see how their theoretically derived g_f (as the composition of f′ and the link function) compares experimentally against the heuristic choice provided in Ref. [34]. Consequences for deep learning and experimental results: Under the assumption that the generator network is reversible, the authors show that there exists an activation function v such that the generator’s hidden layers factor exactly as escorts for the deformed exponential family. In Table 2 (left), the authors compare several generator architectures against different choices of activation functions (that may or may not comply with this escort factorization). The plot for the GAN MLP, for instance, indicates that the “theoretically superior” μ-ReLU performs actually worse than the baseline ReLU (limiting case μ → 1). From their analysis, I would however expect the invertible μ-ReLU (satisfying reversibility assumption) to perform better than the non-invertible baseline (not satisfying reversibility assumption). Can the authors explain why the μ-ReLU performs worse? Can they comment on whether the DCGAN architecture satisfies the reversibility assumption and how these results compare against the Wasserstein GAN which is not based on f-divergences (and for which I therefore do not expect their theoretical analysis to hold)? Finally, as far as I can tell, their results on whether one activation (Table 2 center) or link function (Table 2 right) performs better than the others are all within error bars. I don’t think this provides sufficient support for their theoretical results to be relevant in practice. Minor concerns: Many of their results are derived within the framework of deformed exponential densities. As the general ML community might not yet be familiar with this, I would welcome it if the authors could provide more intuition as to what a deformation (or signature) and an escort is for instance. In Theorem 8, the authors derive several upper bounds for J(Q). How do these bounds compare against each other and what are the implications for which activation functions to use? They also mention that this upper bound on J(Q) is decreasing with the normalization parameter of the escort Z. Can they elaborate a bit more on why this is a good thing and how we can leverage this? In light of these comments, I believe the insights gained from the theoretical analysis are not substantial enough from the point of view of a GAN practitioner.